# Radio emissions reveal Alfvénic activity and electron acceleration prior to substorm onset

S. Y. Wu [1] ✉, D. K. Whiter[1], L. Lamy [2,3], M. M. Wang[4], P. Zarka [3], C. M. Jackman [5], S. Y. Ye[6], J. E. Waters [3], A. R. Fogg [5], S. B. Mende[7], N. Kaweeyanun [1], Y. Kasaba[8], S. Kurita [9], H. Kojima [9], L. J. Paxton [10], Y. Kasahara[11], Y. Miyoshi[12,13], A. Shinbori[12] & F. Tsuchiya[8]

Magnetospheric substorms are among the most dynamic phenomena in the Earth's magnetosphere, yet their triggering mechanisms remain unclear. Ground-based observations have identified wave-like aurora as precursors to substorms. Here, we report a new precursor feature in space-based observations of auroral kilometric radiation (AKR), marked by the appearance of emissions with slowly frequency-drifting tones (<2 kHz/s) above 100 kHz. Simultaneous multi-instrument observations and statistical analysis suggest that these AKR precursors occur concurrently with wave-like aurora, either manifesting as pseudo-breakup features (with about 5 min duration) or as pre-onset activity (about 2 min prior to substorm expansion), indicating a common driving mechanism. Analysis of the emissions with frequency-drifting tones suggests that they are linked to moving double layers driven by dispersive Alfvén waves, consistent with an Alfvénic acceleration mechanism for pre-onset aurora. These findings highlight the importance of Alfvénic activity in substorms and suggest that Alfvénic acceleration is not only responsible for optical auroral features but also for radio emissions, potentially explaining also the ubiquitous frequency-drifting emission features observed at other magnetized planets like Saturn and Jupiter.

Magnetospheric substorms are a key manifestation of interactions between the Earth's magnetosphere and the solar wind, involving energy loading and unloading processes. First characterized from ground-based auroral observations[1], substorms are classically divided into three phases: growth, expansion, and recovery[1,2]. The expansion phase, during which explosive energy release leads to sudden auroral brightening and poleward expansion, remains one of the most debated aspects of substorm dynamics[3-6].

The physical mechanisms underlying substorm onset are broadly categorized into two models. The near-Earth neutral line (NENL) model

[1]School of Physics and Astronomy, University of Southampton, Southampton, UK. [2]Aix Marseille University, CNRS, CNES, LAM, Marseille, France. [3]LIRA, Observatoire de Paris, CNRS, PSL, Sorbonne Université, Université Paris Cité, Meudon, France. [4]Swedish Institute of Space Physics, Uppsala, Sweden. [5]School of Cosmic Physics, DIAS Dunsink Observatory, Dublin Institute for Advanced Studies, Dublin, Ireland. [6]Department of Earth and Space Sciences, Southern University of Science and Technology, Shenzhen, People's Republic of China. [7]Space Science Laboratory, University of California, Berkeley, USA. [8]Planetary Plasma and Atmospheric Research Center, Graduate School of Science, Tohoku University, Aoba, Sendai, Japan. [9]Research Institute for Sustainable Humanosphere, Kyoto University, Uji, Japan. [10]The Johns Hopkins University Applied Physics Laboratory, Laurel, USA. [11]Graduate School of Natural Science and Technology, Kanazawa University, Kanazawa, Japan. [12]Institute for Space-Earth Environmental Research, Nagoya University Graduate School of Engineering, Nagoya University, Nagoya, Japan. [13]Kyung Hee University, Swon, Korea. ✉e-mail: Siyuan.Wu@soton.ac.uk

suggests that magnetic reconnection initiates at distances of approximately 10–20 Earth Radii (Re) downtail in the plasma sheet and propagates earthward[3,5]. Conversely, the current disruption (CD) model proposes that substorm onset is driven by near-Earth instabilities—such as ballooning or cross-tail current instabilities—which disrupt the cross-tail current and trigger expansion from inside-out[4,7,8]. Within the CD framework, structured auroral forms—particularly auroral "beads" that emerge during the growth-to-expansion transition—are interpreted as ionospheric signatures of equatorial plasma instabilities, potentially mediated by kinetic Alfvén wave coupling[9–11]. These auroral structures thus serve as a vital diagnostic of magnetospheric instability and energy transfer.

Pre-onset auroral activities, especially auroral beads, have received increasing attention due to their consistent appearance prior to substorm onset[11–14]. These structures typically exhibit azimuthal periodicity with wavelengths <~100 km, and statistical studies report their presence in over 90% of substorm events[14]. Theoretical and simulation works have attributed their formation to various plasma instabilities, including interchange ballooning modes and cross-field current instabilities[4,14–16]. Notably, auroral bead properties—velocity, period, and wavelength—align closely with dispersive Alfvén waves (DAWs) in the plasma sheet[10]. Conjugate satellite and ground-based measurements further support this interpretation, showing co-located signatures of auroral beads and DAWs across magnetospheric latitudes[17,18]. Ground-based magnetometer data also detect Alfvénic wave power near the region of aurora beads, with typical wave periods of 10–100 seconds that map to the inner magnetotail (~10 Re)[11,19]. These findings collectively suggest that DAWs play a critical role in the development of auroral beads.

While the term auroral beads is widely used, these structures display diverse morphologies depending on the observational context. Previous studies have referred to them as "ray-like"[20], "spot-like"[13,21], or more generally as "wave-like" forms[22]. These structures can evolve azimuthally into twisted or vortex-like patterns[23]. To accommodate this diversity in morphology and evolution, we adopt the term wave-like aurora forms (WAs) to refer to these auroras that are observed exactly before substorm onset. Note that WAs are not limited to substorm events but are also observed during pseudo-breakups[14] and magnetically quiet periods[24], suggesting a broader relevance to magnetosphere–ionosphere coupling.

In addition to optical auroral activity during substorms, radio-frequency emissions above the atmosphere, specifically Auroral Kilometric Radiation (AKR), show a strong correlation with substorm events[25–27]. AKR enhancements are often detected 1–2 min before auroral breakups[28]. Detailed studies have identified a two-step AKR evolution during substorms: an initial enhancement of high-frequency emissions coinciding with auroral brightening, followed by a more intense enhancement and low-frequency extension associated with auroral breakup and poleward expansion[29–31]. This early AKR enhancement is recognized as a precursor to substorm onset, supported by both case studies[29,30] and superposed epoch analyses[32], with emissions observed up to 20 min before onset[32]. These AKR precursors are believed to originate from large-scale inverted-V acceleration regions or electrostatic structures, such as double layers[31,33].

AKR is produced through the cyclotron maser instability by aurora electrons precipitating along magnetic field lines with energies of a few keV[34], linking AKR directly to auroral activities[25,35–37]. The emissions are generated with frequencies close to the electron cyclotron frequency in the source region and beamed nearly perpendicular to the source magnetic field lines[38,39].

AKR has been associated with a variety of auroral structures, such as discrete arcs[25], substorm expansion bulges[35], and transpolar arcs[36]. While AKR is traditionally attributed to electrons accelerated by quasi-static potential drops[33,40], Alfvénic acceleration has also been implicated in certain AKR forms[41–45]. Given that both AKR precursors and

WAs serve as indicators of substorm onset, and that AKR is intrinsically linked to auroral dynamics, examining the connection between these phenomena is of significant interest.

Here, we show that AKR precursors occur during both pseudo-breakup intervals and pre-onset phases, and that they are connected to WAs through simultaneous observations and statistical analysis. Results suggest that these precursors represent the radio-frequency counterparts of WAs, appearing as repetitive frequency-drifting tones generated by DAWs driving double layers along auroral field lines. This mechanism provides a universal pathway for producing frequency-drifting radio emissions across planetary magnetospheres, from Earth to Saturn and potentially Jupiter.

## Results

### Auroral structures preceding substorm onset

Figure 1 presents a sequence of auroral observations from a ground-based all-sky imager, capturing the evolution of auroral structures prior to substorm onset. The onset is identified at ~23:25:20 UT, marked by a clear poleward expansion of auroral emissions (Fig. 1a), a gradual increase in the AE index (Fig. 1b), and a moderate negative excursion in the AL index (Fig. 1c).

Approximately 5 min before onset (23:19:40 UT), a group of azimuthally periodic auroral structures appears in the camera's FOV, propagating eastward. These structures intensify and reach peak brightness around 23:22:20 UT, then gradually decay. Initially exhibiting well-defined "bead-like" forms, they become less distinct over time, evolving into more continuous "wave-like" aurora. At ~23:24:00 UT, a second group of auroral structures with similar wave-like morphology emerges and continues intensifying up to the moment of onset, coinciding with the start of poleward expansion.

This observational sequence—an initial set of periodic auroral brightening without poleward expansion, followed by a second structured auroral form leading into substorm onset—is frequently observed. The first group can be classified as a pseudo-breakup or pseudo-storm[46,47], representing localized auroral activity that does not lead to a full substorm. The second group corresponds to the actual substorm onset. It is important to note that pseudo-storm aurora and substorm onset aurora do not always occur sequentially. They may develop independently or in close succession and may appear along either the same pre-existing auroral arc or newly formed arcs[46,48]. Morphologically, both types often exhibit azimuthally periodic structures—either as bead-like forms (more discrete in shape), more continuous wave-like arcs, or as an evolution from bead-like to wave-like patterns.

### AKR emissions preceding substorm onset

Figure 2 presents five representative examples of AKR precursor emissions observed around substorm onset, identified using data from the plasma wave investigation (PWI) instrument onboard the Polar spacecraft[49]. These events were selected based on previously established substorm onset times, determined independently from ground-based magnetometers or global auroral images[50–53] (see Methods). In each panel, the substorm onset time is marked by a blue dot-dashed line, while the white dot-dashed line denotes the AKR "breakup" time, defined as the onset of intense AKR emission and low-frequency extension. These AKR-derived onset timings generally align well with those determined from optical and magnetic field measurements.

Between the two white lines in each panel, the AKR precursor emissions appear as faint enhancements or abrupt signal onsets emerging from a quiet background. Although weaker and more spectrally confined than the AKR during the expansion phase, these emissions are consistently detectable. Importantly, we report for the first time that many precursors exhibit multiple frequency-drifting tones with a quasi-periodic or repetitive structure, most clearly seen in Fig. 2a–d. In some events (e.g., Fig. 2e), the precursor emissions occur

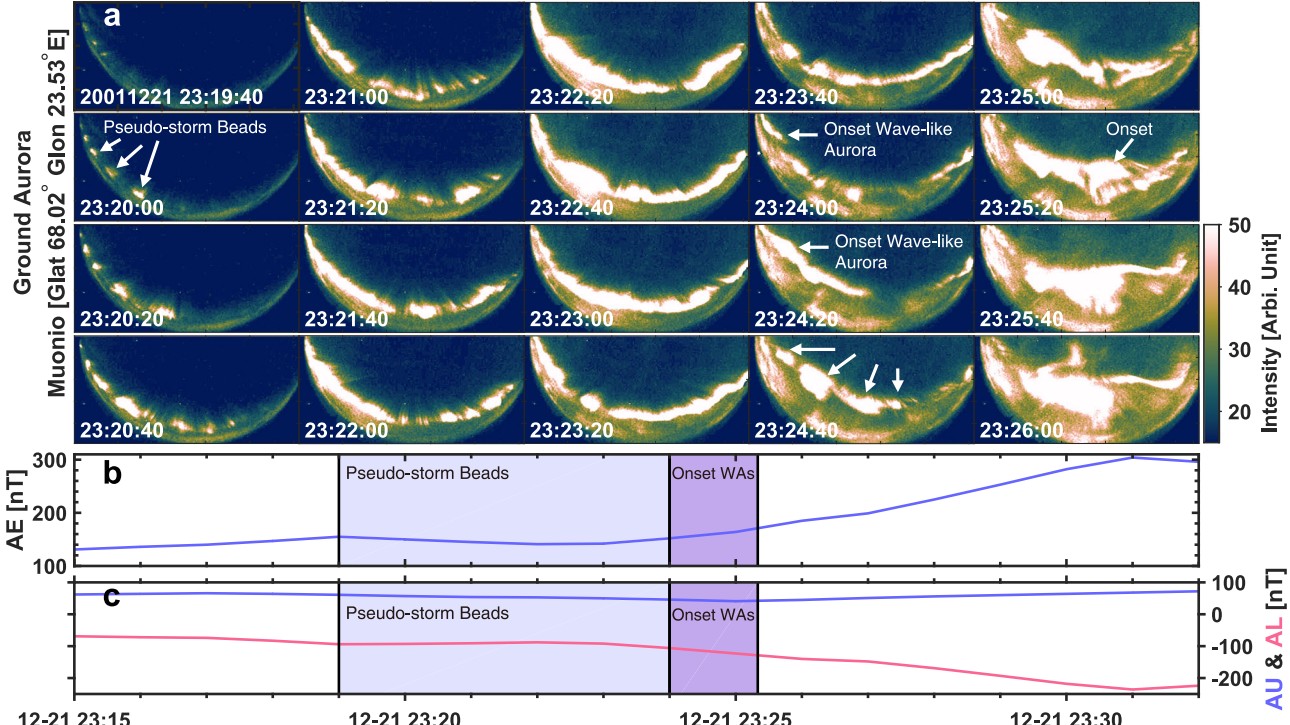

**Fig. 1 | Evolution of auroral activity prior to substorm onset. a** Ground-based all-sky camera observations of auroral evolution recorded at Muonio (geographic latitude 68.02° N, longitude 23.53° E) using a 557.7 nm filter. **b, c** Geomagnetic indices (AE, AU, AL) showing the corresponding magnetospheric activity. The light blue shaded interval marks the pseudo-storm period preceding the substorm. The light purple shaded region denotes the interval of wave-like auroral activity immediately preceding the substorm onset, which is identified at about 23:25:20 based on the initiation of auroral poleward expansion. Source data are provided as a Source Data file.

in several distinct groups with similar spectral characteristics, some of which are associated with pseudo-storm activity, as further supported by concurrent geomagnetic indices (Fig. 2f, g). This pattern suggests a potential link between these radio features and pre-onset or pseudo-breakup auroral activity, closely mirroring the auroral evolution in Fig. 1, where an initial pseudo-storm is followed by a new set of auroral structures that culminate in a full substorm onset.

A comprehensive survey of Polar observations between March 1996 and September 1997 allowed us to identify 266 cases of substorm-related AKR precursors (see Methods). Among these, 186 events showed clear frequency-drifting tones—defined as events with at least one discernible drifting feature—while the remaining 80 cases exhibited smoother structures, likely due to stronger background emissions or limitations in the time-frequency resolution (see Supplementary Data 1). An additional 93 cases were identified and associated with pseudo-storms, 71 of which displayed frequency-drifting features. Across both categories, the majority of frequency-drifting tones exhibit positive frequency drifts, implying an upward spectral shift versus time. This behavior is analyzed in more detail in the subsequent sections.

Analysis of the spacecraft position when the AKR precursors are observed reveals that they are predominantly visible at high magnetic latitudes (MLat >30°), on the nightside, and at radial distances greater than 4 Re (see Supplementary Fig. 1). No significant differences in spatial distribution are found between substorm-related and pseudo-storm-related events.

One possible explanation for observing frequency-drifting tones is the spacecraft's motion through spatially structured AKR beams, analogous to drifting features in Jupiter's decametric emissions[54,55]. In this case, the observed drift rates and tone morphology would vary with spacecraft position relative to the beam. To test this, we examined four events with simultaneous observations from the Polar and Geotail spacecraft (Supplementary Fig. 2), all of which revealed consistent frequency-drifting features across both platforms despite their spatial separation. Note that in all four events, the two spacecraft were not located on opposite sides of Earth (e.g., dayside versus nightside as illustrated in Supplementary Fig. 2), which would have prevented them from detecting the same source. Instead, the maximum separation between the two spacecraft was approximately 9 h in magnetic local time (MLT) and 65° in MLat—well within a shared visibility region. Theoretically, the maximum possible MLat separation could be up to 180°, and the maximum MLT difference up to 12 h, assuming a simple 90° beaming angle. This moderate separation supports the conclusion that both spacecraft observed emissions from the same source region. These observations suggest that the frequency-drifting tones observed in AKR substorm precursors are intrinsic to the source dynamics, rather than artifacts of spacecraft motion[54,55].

## Simultaneous observations of AKR precursors and wave-like auroral structures

The above two sections revealed strikingly similar temporal evolutions of WAs and AKR precursors prior to substorm onset, suggesting a common underlying driver. To illustrate this potential connection, we present two simultaneous observations of WAs and AKR precursors.

As shown in Fig. 3a, AKR precursors are clearly identifiable on the wave spectrogram measured by the Plasma Wave Experiment (PWE) instrument onboard the Arase satellite[56–59], marked by white and yellow arrows. These emissions exhibit multiple frequency-drifting tones. The horizontal white dashed line indicates the substorm onset, determined from the poleward expansion of auroral structures observed in the ground-based aurora keogram (Fig. 3b), a time-versus-latitude plot generated from a sequence of all-sky images in Fig. 3c. A striking temporal correlation is evident between the AKR precursors and auroral activity: the timing of the drifting emissions closely aligns with the evolution of the pre-onset auroral arc (indicated by white arrows), which is also appreciable in the auroral images from Gillam

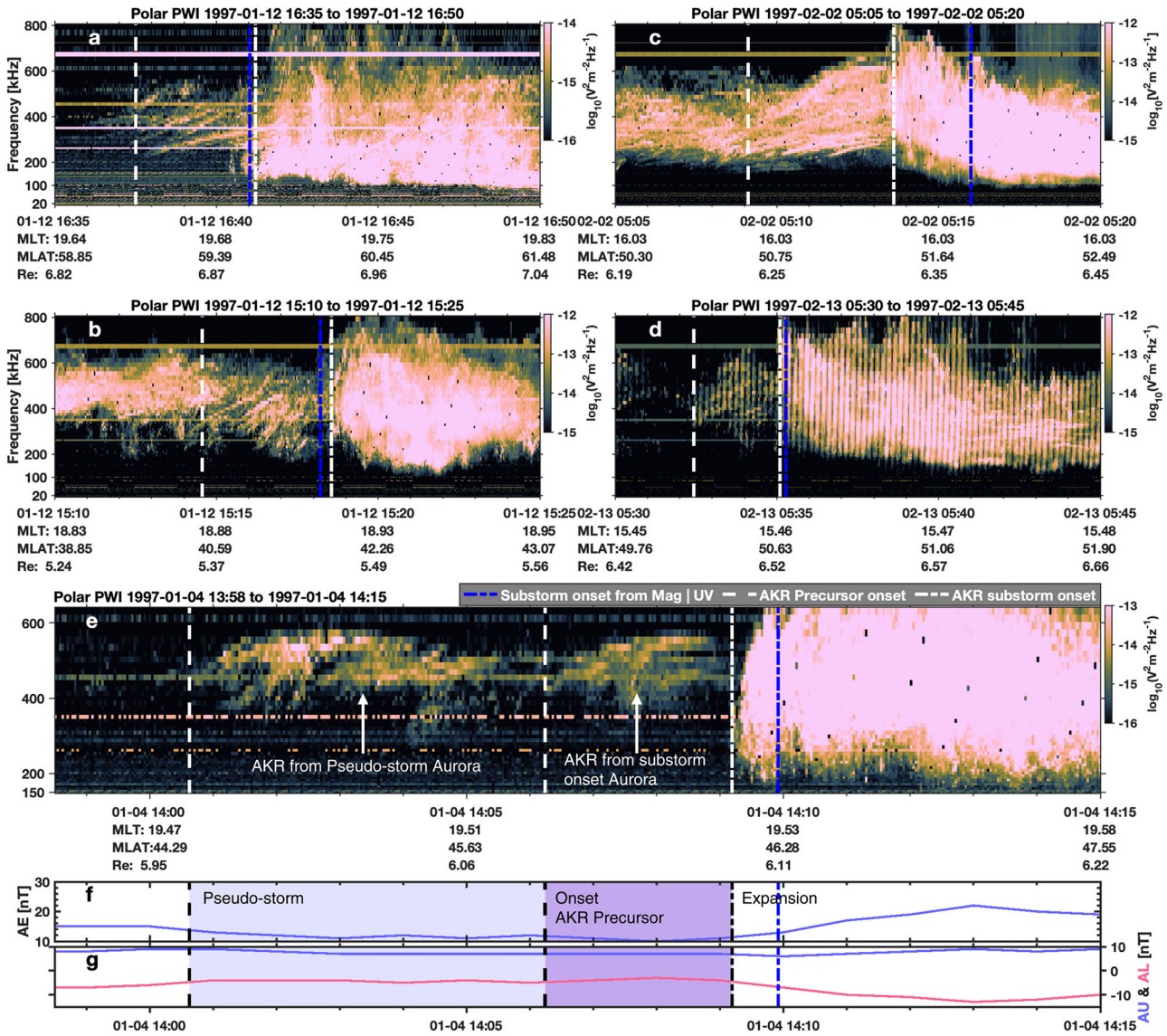

**Fig. 2 | Examples of AKR precursor emissions. a** Dynamic spectrum of AKR precursor emission observed by the Polar Plasma Wave Investigation instrument. The spectrogram shows the radio wave electric field intensity as a function of frequency and time, with color indicating signal strength. The white dashed line marks the onset of the AKR precursor, while the white dot-dashed line indicates the AKR breakup time, corresponding to the substorm onset identified from AKR observations. The blue dot-dashed line denotes the substorm onset time derived independently from ground-based magnetometer data or global auroral imaging, typically coinciding with the AKR breakup time. **b**–**d** Additional examples of AKR precursor emissions observed in separate substorm events. **e** A representative case showing two distinct groups of AKR precursors associated with both a pseudo-storm and a regular substorm. **f**, **g** Geomagnetic indices (AE, AU, AL) during the same interval as shown in (**e**), illustrating the magnetospheric activity associated with the AKR observation. Source data are provided as a Source Data file.

station (Fig. 3c). The arc itself exhibits wave-like features during its evolution. At ~07:31 UT—about one minute before substorm onset—the auroral arc undergoes sudden brightening and equatorward motion. Simultaneously, a new WA structure, consisting of a series of auroral spots (Fig. 3c, yellow arrows), enters the FOV. Correspondingly, the AKR spectrogram reveals a new set of frequency-drifting emissions (highlighted by the yellow arrow on the right side of Fig. 3a).

During this interval, the Arase spacecraft was located near 30° MLat in the northern hemisphere. To verify whether the AKR sources associated with the observed aurora were within Arase's radio field of view (FOV), we performed a radio visibility analysis (see Methods for details), as presented in Fig. 3d. Assuming that AKR emissions are beamed at angles <90° from the local magnetic field direction[60,61], the calculation shows that Arase was well positioned to receive emissions from source regions near its central longitude with smaller beaming angle (e.g., 40° for these yellow-brown color markers). As the beaming angle increases, the observable longitudes expand symmetrically around the spacecraft's longitude. The projected visible AKR source footprints in magnetic polar coordinates are shown in Fig. 3e. The background global auroral image was captured by the global ultra-violet imager (GUVI) instrument onboard the TIMED spacecraft[62]. These results indicate that the emissions associated with the auroral activity seen at Gillam could have propagated to the Arase spacecraft with beaming angles between ~80° and 90°. This confirms that Arase had visibility to the AKR sources responsible for the observed precursor emissions, thereby supporting the spatial and temporal association between WAs and AKR precursors.

Figure 4 presents a second event demonstrating the simultaneous occurrence of AKR precursor and WAs. The wave electric field spectrogram (Fig. 4a), obtained by the plasma wave instrument (PWI) onboard Geotail[63], reveals AKR precursors (marked by white, yellow, and pink arrows), with emissions beginning around 20:30:00 UT and

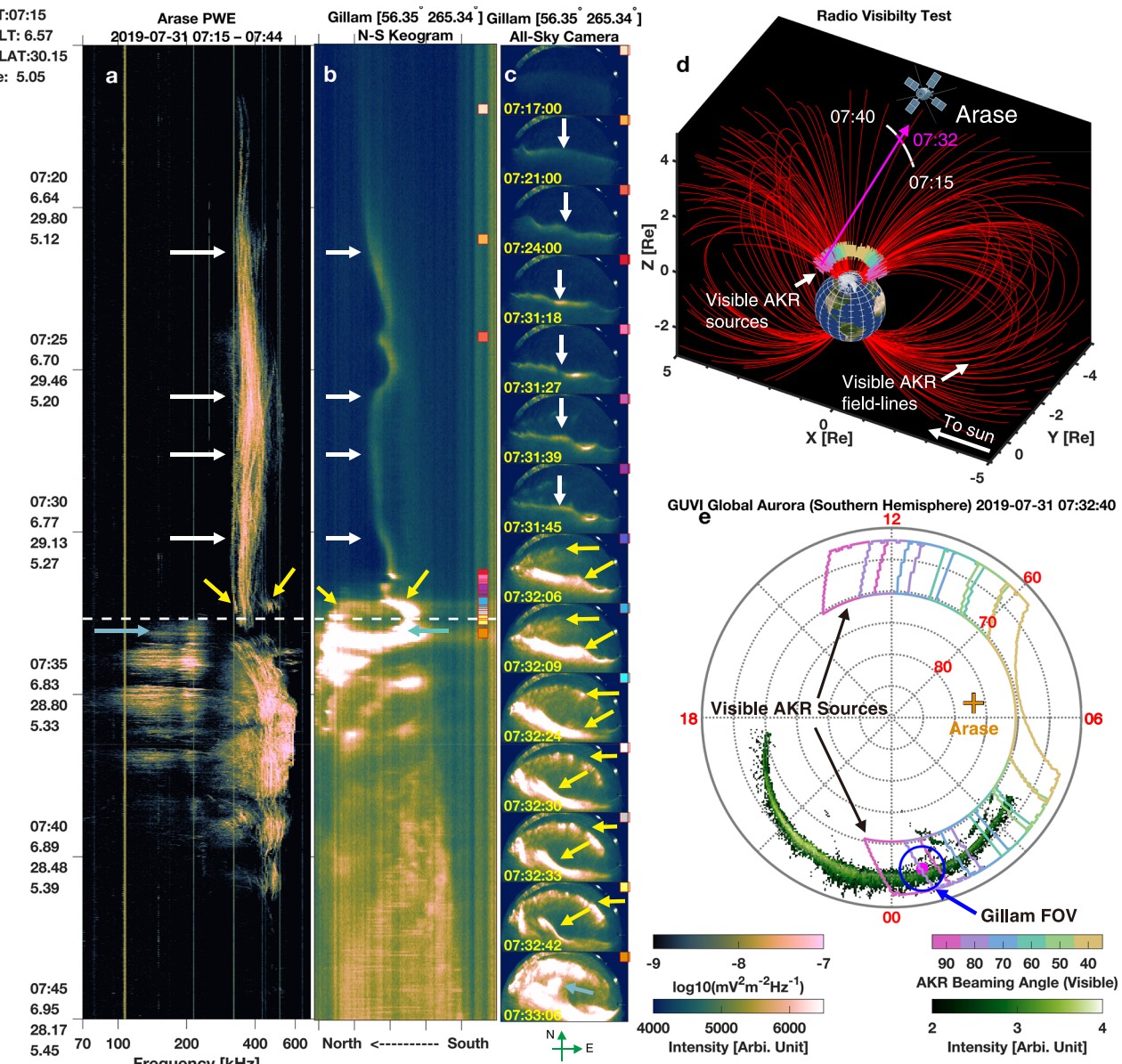

**Fig. 3 | Simultaneous observations of AKR precursors and wave-like auroral forms: Case 1. a** Wave electric field spectrogram measured by the plasma wave experiment instrument onboard the Arase satellite from 07:15:00 to 07:45:00 UT on 31 July 2019. AKR precursors are indicated by the white and yellow arrows. The substorm onset time (white dashed line) is determined from the poleward expansion of aurora in (**b**). Vertical stripes on the spectrogram result from instrumental noise caused by onboard systems. Colored arrows across (**a**–**c**) highlight temporally and spatially correlated AKR emissions and auroral structures. **b** North–south keogram derived from ground-based auroral images captured by the all-sky imager at Gillam station (geographic latitude 56.35° N, longitude 265.34° E), using a 557.7 nm filter. **c** Selected auroral images from the Gillam all-sky camera. Colored squares in the top-right corners of each image indicate their respective timestamps in the keogram shown in (**b**). **d** Radio visibility map during the observation interval in (**a**). The white line denotes the Arase trajectory, with the pink segment corresponding to the time range in (**a**). Visible AKR source locations are color-coded by radio beaming angle, as indicated in the bottom-right colorbar. Red field lines represent magnetic field lines connecting visible AKR sources. **e** Footprints of visible AKR sources projected onto a magnetic polar coordinate system. The background image shows the global aurora in green–white colormap, observed by the GUVI instrument onboard the TIMED satellite. Colored solid lines delineate the boundaries of AKR source footprints for different beaming angles. The pink spot and blue circle indicate the location and field of view of the Gillam all-sky camera. The orange cross marks the Arase footprint at 07:32:40 UT. Source data are provided as a Source Data file.

the substorm onset timing identified at ~20:40:00 UT. Although these precursors do not exhibit prominent frequency-drifting features—likely due to the limited time-frequency resolution of the Geotail instrument—a slight positive drift is discernible around 20:32 UT before it merges into broader background emissions.

Ground-based auroral observations from the Abisko station (Fig. 4b) reveal the evolution of WAs, beginning around 20:31:00 UT—nearly simultaneous with the AKR precursor onset. These auroral structures propagate eastward and reach peak intensity near 20:34:00

UT. The intensity then diminishes, followed by the emergence of a second group of WAs at about 20:38:20 UT, with auroral breakup occurring ~20:40:00 UT. This sequence aligns with the AKR observations, where emissions similarly fade (orange arrows) and then intensify again (pink arrows). This sequence parallels the scenario presented in Figs. 1 and 2e, where the first WA structure corresponds to a pseudo-storm, and the second to the substorm onset. Although frequency-drifting features are unresolved in this case, the temporal evolution is consistent with that of the aurora.

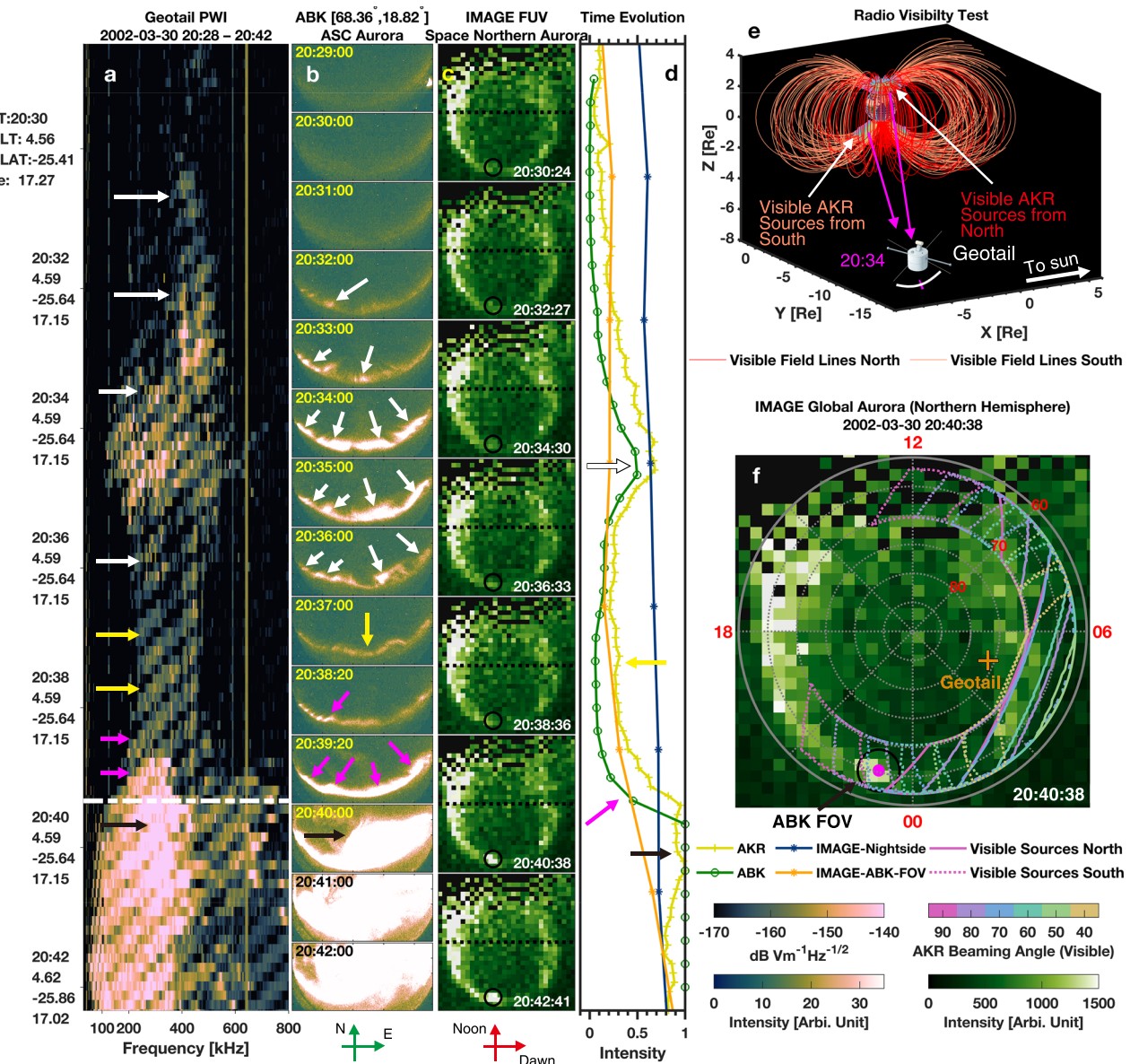

**Fig. 4 | Simultaneous observation of AKR precursors and wave-like auroral forms: Case 2. a** Wave electric field spectrogram measured by the Geotail Plasma Wave instrument from 2002-03-30 20:28:30 to 20:42:20 UT. AKR precursor emissions are indicated by white, yellow, and pink arrows. Negatively drifting features are instrumental artefacts related to the frequency-sweeping logic of the receiver. **b** Ground-based auroral images captured at 557.7 nm by the all-sky camera at the Abisko (ABK) station. **c** Global proton auroral images from the Spectrographic Imager (S12) onboard the IMAGE satellite. The black circle marks the field of view of the Abisko station. The horizontal black dashed line denotes the dawn–dusk meridian. **d** Normalized integrated intensities of: AKR emissions (yellow; 100–800 kHz from **a** -8 s resolution), ground-based aurora (green; integrated over the entire

image in (**b**), 20 s resolution), global nightside aurora (dark blue; integrated below the dashed line in (**c**)), and local aurora within the Abisko field of view in (**c**) (orange; 2 min 3 s resolution). Normalization follows: $\frac{\text{integrated} - I_{min}}{I_{max} - I_{min}}$. For saturated ground-based images after onset, a denominator of $0.6 \times (I_{max} - I_{min})$ is applied empirically. **e** Radio visibility test for the interval shown in (**a**). Visible AKR source field lines from the Northern hemisphere are shown in red; those from the Southern hemisphere in orange. **f** Footprints of visible AKR sources projected onto magnetic polar coordinates. The background is the same IMAGE auroral frame as in the 2nd bottom row of (**c**). Boundaries of northern visible sources are shown as solid colored lines; southern hemisphere sources as dotted colored lines. Source data are provided as a Source Data file.

Global proton aurora images from the IMAGE satellite[64] (Fig. 4c) confirm that Abisko station was located directly beneath the substorm onset region, with the FOV marked by a black circle. Notably, AKR precursors precede the visible auroral features by about one minute, possibly due to the initial auroral activity occurring outside the camera's FOV.

Figure 4d presents normalized integrated intensity profiles of AKR (yellow), ground aurora (green), global nightside aurora (dark blue), and the local aurora within Abisko's FOV from IMAGE data (orange). The AKR and ground-based aurora exhibit strongly correlated variations, while the global nightside aurora shows no distinct

correspondence. The local aurora signal extracted from IMAGE increases around substorm onset but lacks clear peaks during the WAs evolution phase, likely due to insufficient spatial resolution. These findings reinforce the interpretation that AKR precursors originate along magnetic field lines connected to WAs during both pseudo-storm and pre-onset intervals.

At the time of observation, Geotail was located in the southern hemisphere at a magnetic latitude of −25°, with an -6-hour magnetic local time difference from Abisko. This geometry suggests that the observed emissions may have been beamed from either the conjugate WAs in the southern hemisphere or from the northern hemisphere

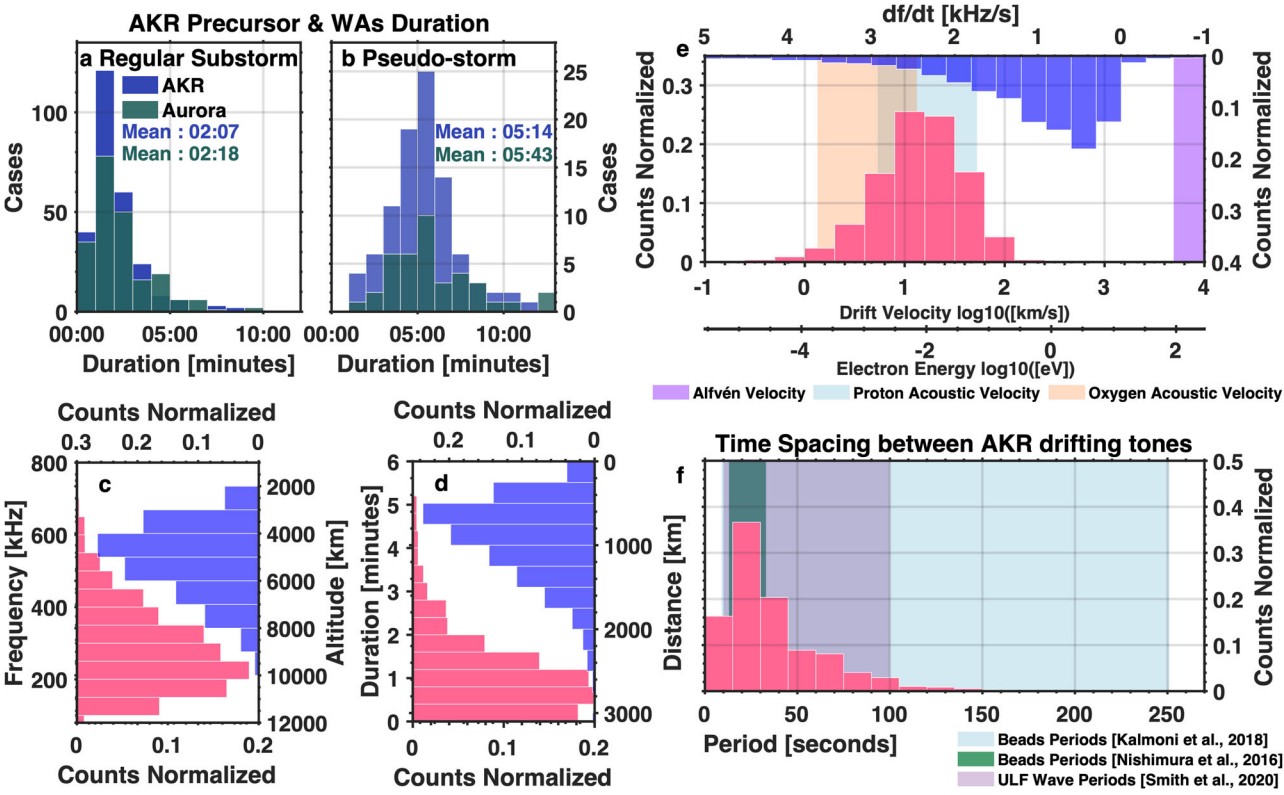

**Fig. 5 | Statistical properties of AKR precursors and wave-like aurora forms.**
**a** Time duration distributions of AKR precursors (blue) and wave-like aurora (green) associated with regular substorms. **b** Time duration distributions of AKR precursors (blue) and wave-like aurora (green) associated with pseudo-storms. **c** Frequency and inferred source altitude distribution of AKR precursors (including both substorm and pseudo-storm events). The red histogram shows the frequency distribution of the precursors, while the blue histogram presents the corresponding source altitudes, derived by tracing magnetic field lines from typical wave-like aurora form locations using the IGRF model. **d** Duration and propagation distance of frequency-drifting tones along magnetic field lines (substorm and pseudo-storm events combined). The red bars denote the durations, and the blue bars represent estimated travel distances. **e** Frequency drift rates (df/dt) of the precursor tones (blue) and the corresponding inferred drift velocities (red), assuming the source motion along magnetic field lines. Shaded background regions indicate characteristic velocities in the source region: Alfvén speed (purple), proton acoustic speed (light blue), and oxygen ion acoustic speed (light orange), as estimated from plasma parameters described in the Methods. **f** Periodicity of repetitive frequency-drifting tones in AKR precursors (substorm and pseudo-storm events combined). The red histogram indicates the observed periodicities. Shaded regions show the typical period ranges of auroral beads from previous studies (blue[14], green[20]) and ultra-low frequency (ULF) waves associated with auroral beads (purple[11]), for comparison. Source data are provided as a Source Data file.

with a wide beaming angle. AKR and WAs are known to be magnetically conjugate and often display symmetric responses to substorm activity[65,66]. The radio visibility test in Fig. 4e, f supports this interpretation, showing that a beaming angle of ~70–80° from a conjugate southern hemisphere source would allow Geotail to detect the emissions. By contrast, a northern hemisphere source would require an implausibly large beaming angle (~90° or more). Note that, unlike in Fig. 4, no calculation of visible sources from the southern hemisphere is performed in Fig. 3, as Arase was located at a smaller radial distance where the dense plasmasphere would obstruct emissions from the opposite hemisphere[67]. An additional case demonstrating clearer simultaneity and more favorable spacecraft viewing geometry is presented in Supplementary Fig. 3.

**Characteristics of AKR precursors and wave-like auroral forms**
To statistically examine the relationship between AKR precursors and WAs, we analyzed 142 events identified from all-sky auroral images recorded between December 1996 and December 2007 by the MIRACLE camera network[68]. These events were selected based on the appearance of azimuthally WA forms before substorm onsets. Among them, 103 events were associated with regular substorms and 39 with pseudo-storm activity. In addition, we included 111 previously reported auroral bead events associated with regular substorms from an established database[20]. This combined dataset enables a robust comparison of AKR precursor characteristics with WAs across a broad set of events.

We first compared the duration of WAs and AKR precursors. The duration of WAs was measured from the time of their initial appearance in all-sky camera images until auroral breakup (for substorms) or fading (for pseudo-storms). AKR precursor duration was defined from their appearance on radio spectrograms until either the AKR breakup (for substorms) or the end of frequency-drifting emissions (for pseudo-storms). As shown in Fig. 5a, b, both phenomena exhibit comparable durations: approximately 2 min prior to substorm onset for substorm-related cases, and ~5–6 min for pseudo-storms. Despite differences in duration between substorm and pseudo-storm events, the spectral characteristics of the frequency-drifting emissions are remarkably similar. Therefore, we combined both categories for subsequent analysis. A detailed comparison between the two subsets is provided in the Supplementary material (Supplementary Figs. 4 and 5).

The frequency distribution of AKR precursors (Fig. 5c) spans 100–700 kHz, with an average frequency ~283 kHz. Based on the assumption that AKR is generated at the local electron cyclotron frequency[38,39], we used the IGRF magnetic field model[69] to map these frequencies to source altitudes. The resulting source altitudes range from ~2000 km to 8,000 km (0.31–1.25 Re), with a mean altitude ~5100 km (0.8 Re).

We identified and extracted frequency-drifting tones from AKR spectrograms for further analysis. These tones have an average duration of 1 min and 7 seconds (Fig. 5d), and each precursor event typically contains multiple, distinct tones. Since AKR frequency corresponds to

the local electron cyclotron frequency, observed frequency drifts imply changes in the source altitude or magnetic field strength. Assuming source motion along magnetic field lines, the estimated travel distance is ~947 km (0.15 Re). The average drift speed, derived from the frequency drift rates (typically <3 kHz/s), is ~21 km/s (Fig. 5e). Around 96% of identified tones exhibit positive frequency drifts, suggesting motion from higher to lower altitudes—toward regions of stronger magnetic field. We note, however, that in rare cases—~4% of the events—AKR precursors exhibit negative frequency drifts, indicating source motion toward higher altitudes, away from the Earth. The physical conditions that determine whether the source drifts upward or downward remain unclear and require further investigation.

To explore the origin of the frequency-drifting tones, we compared their drift velocities with characteristic velocities in the AKR source region. The calculated electron kinetic energies, derived from the drift velocities, range from $10^{-4}$ to $10^{-1}$ eV (see secondary $x$ axis in Fig. 5e), which is much lower than the keV-range energies typically associated with AKR source electrons[39,54]. This rules out the free motion of AKR source electrons along magnetic field lines as the cause of the drift, as previously reported for Jupiter's radio emissions[55,70]. The calculated Alfvén speed (Fig. 5d, purple shaded) is also higher than the drift velocities, excluding a direct connection to Alfvén waves. These velocities also differ from those of ion holes driving striated AKR bursts[71]. In contrast, the ion acoustic speeds (Fig. 5d, blue and light orange shaded) align well with the observed drift velocities, suggesting that the frequency-drifting tones are generated by AKR sources moving along magnetic field lines at ion acoustic speeds.

Many of the drifting tones also exhibit periodic structures (e.g., Fig. 1b). The recurrence period of these tones, typically less than two minutes and averaging 37 seconds (Fig. 5f), closely matches the characteristic periods of WAs reported in previous studies[10,20]. This periodicity is also consistent with that of DAWs and ultra-low frequency wave activity observed during substorm growth phases[10,11], reinforcing the link between AKR precursors and WAs. This consistency across phenomena suggests a unified physical mechanism. The periodic frequency-drifting tones in AKR precursors are also likely driven by DAWs, which stimulate AKR sources moving along magnetic field lines at local ion acoustic speeds.

The periodicity of WAs arises from their spatial wavelength and azimuthal motion, producing a periodic pattern for a stationary observer. When mapped to the magnetosphere, this wavelength and period align with DAWs. Similarly, the periodicity in AKR precursors may result from the azimuthal motion of WAs, with connected AKR radio sources exhibiting a spatially separated feature and behaving as periodic beams sweeping across the spacecraft. Alternatively, these tones could stem from the periodic formation of AKR sources as DAWs pass through the radio visible region. Distinguishing between these temporal and spatial effects is challenging and requires further investigation.

## Discussion

The frequency-drifting emissions in AKR precursors closely resemble the fine structures previously identified in AKR, with drift rates of several kHz/s[72,73]. These emissions are linked to double layers moving along magnetic field lines at ion acoustic speeds[33,72,73], consistent with our observations. However, the exact mechanism driving double-layer formation—whether through quasi-static potential drops or DAWs—remains unclear. We propose that DAWs play a key role in stimulating double-layer formation, which subsequently generates the observed frequency-drifting emissions. This is particularly evident in the periodic tones associated with AKR precursors, whose periods align with those of DAWs and are frequently detected before substorm onset. The positive frequency drifts in these precursors support earlier findings that DAWs, generated around 10 Re in the magnetosphere, propagate toward the auroral acceleration region[9–11]. Upon reaching

this region, these DAWs stimulate double layers, producing the observed emissions.

The frequency-drifting emissions, with drift rates of several kHz/s, are not unique to AKR precursors but represent a ubiquitous phenomenon, occurring in up to 20% of high-resolution AKR observations[73]. Their prominence in low-resolution spectrograms prior to substorm onset suggests enhanced Alfvén wave activity during these periods. Note that these AKR fine structures with similar frequency-drifting characteristics observed outside precursor phases may also correspond to certain auroral forms with Alfvénic signatures, e.g., the non-substorm "beads-like" auroras as previously reported[24]. While this connection lies beyond the scope of the present study, it merits further exploration.

Similar frequency-drifting structures have also been identified in Saturn's kilometric radiation, with drifting rates ranging from a few kHz/s to as high as 240 kHz/s[74]. At Jupiter, periodic radio sources moving at local ion acoustic speeds have been observed in radio source regions[75,76]. These observations infer small-scale electric potential structures that strongly suggest the presence of double layers. Moreover, the role of Alfvén waves in generating radio emissions at Jupiter is even more prominent and well established—particularly in satellite-induced emissions[54,70,75,76]. These findings suggest the idea that the mechanism proposed in this study—where frequency-drifting AKR emissions are driven by double layers formed through DAW interaction—may also apply to Saturn and Jupiter. This points to a potentially universal process responsible for frequency-drifting radio emissions across magnetized planetary environments.

It is important to note that the AKR precursor frequency-drifting tones discussed here should not be confused with previously reported fine structures of AKR, such as striated AKR[71]—characterized by rapid negative frequency drifts (about 5 kHz/s), typically observed at frequencies below ~90 kHz—and S-burst-like AKR emissions[44,45], which exhibit even faster drift rates (>10 kHz/s), shorter durations, and recurrence intervals of ~7–18 kHz. Additionally, "dot" AKR—appearing as spot-like features in AKR spectrograms and predominantly observed below 100 kHz—has also been reported[43]. Although both S-burst-like and dot AKR emissions have been proposed to be associated with Alfvén wave activity, they are generally attributed to direct Alfvénic acceleration without invoking the formation of double layers. A further key distinction lies in their contextual association: the fine structures reported in earlier studies have not been systematically linked to specific magnetospheric processes and are more likely to represent ubiquitous features of AKR. In contrast, the AKR precursors identified in this study are consistently observed immediately prior to substorm onset or during pseudo-substorm auroral activity, indicating a different origin and potentially distinct generation mechanism. This highlights a previously underexplored connection between AKR fine structures and large-scale magnetospheric dynamics, particularly in the context of substorm triggering.

AKR is typically associated with quasi-static potential drops and inverted-V electron precipitations[31,39], although DAWs are also proposed to contribute to certain forms of AKR[41–45]. The processes responsible for AKR generation, as well as the creation of CMI-unstable electron distributions, extend beyond potential drop and Alfvénic acceleration alone. Our findings suggest that DAWs not only accelerate electrons but also pump energy to sustain double layers, ultimately generating frequency-drifting AKR emissions. Therefore, DAWs may directly contribute to the generation of certain AKR forms, while also indirectly exciting AKR through the formation of double layers. The formation of double layers by DAWs has been both predicted in theoretical simulations[6,77,78] and demonstrated by in-situ observations[79,80], although this phenomenon has received limited attention in the context of AKR and substorms. The whole scenario of the generation of WAs and AKR precursors is given in Fig. 6.

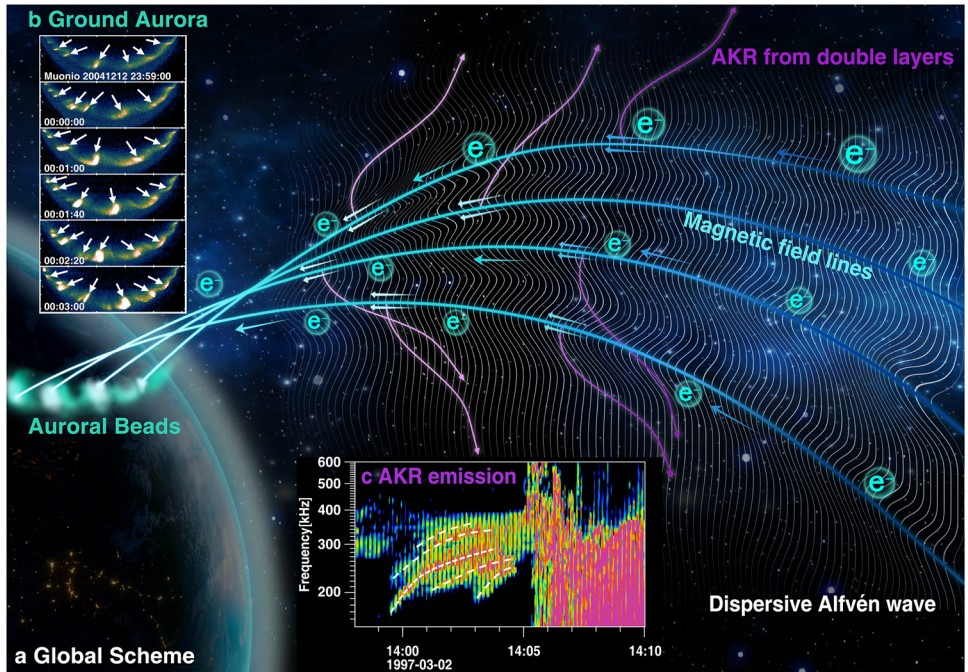

**Fig. 6 | Schematic illustration of the generation mechanism for AKR precursors and auroral beads (Wave-like aurora). a** Overview of the global process: dispersive Alfvén waves (white background wavy lines) accelerate electrons, stimulating double-layer structures (two parallel arrowed lines along magnetic field lines). These accelerated electrons precipitate along magnetic field lines, producing auroral beads in the ionosphere. Simultaneously, the earthward movement of the double layers generates the frequency-drifting AKR emissions. The bead's wavelength plotted in the ionosphere roughly corresponds to the spatial wavelength of the DAWs in the magnetosphere according to the previous study. **b** Example of auroral beads observations captured at the Muonio station. **c** Typical AKR precursor observations from the Polar PWI instrument, showing frequency-drifting tones indicated by the white dashed lines. Source data are provided as a Source Data file.

The simultaneous observation of AKR precursors and WAs provides compelling evidence for enhanced DAWs activity preceding substorm onset. This finding not only supports previous studies suggesting the Alfvénic nature of pre-onset auroras[81] and auroral beads[9,20], but also highlights a more indirect role of Alfvén waves in these dynamics, such as their involvement in generating double layers. Recent studies propose that double layers, formed by DAWs and the resulting feedback loops, may trigger substorm expansion onset[6].

In summary, this study characterizes AKR precursors observed during both pseudo-breakup intervals and pre-onset phases, and establishes their connection to WAs through a combination of simultaneous observations and statistical analysis. AKR precursors serve as radio-frequency counterparts to WAs, occurring at higher altitudes along magnetic field lines linked to the WAs. These precursors present as repetitive frequency-drifting tones, consistent with a mechanism where DAWs stimulate double layers moving along auroral field lines at ion acoustic speeds. This mechanism may represent a universal mechanism for generating frequency-drifting radio emissions observed in multiple planetary magnetospheres, from Earth to Saturn and potentially Jupiter, which underscores a fundamental role of Alfvénic acceleration in the generation of planetary radio emissions across diverse magnetized environments.

## Methods

### Identification of AKR precursors and WAs
AKR precursors were identified through a detailed examination of radio emission data collected around pre-determined substorm onset times. These onset times were derived from two primary sources: ground-based magnetometer observations[52,53] and global auroral images[50,51]. Substorm onset times from ground-based magnetometer data were identified using automated detection algorithms applied to the SuperMag AL (SML) index[52,53]. This method leverages extensive magnetometer networks to pinpoint sudden decreases in the SML index, which are indicative of substorm activity. Substorm onsets were also determined through automated detection of auroral brightening and poleward expansion in global auroral images. This approach utilized global aurora images from both the Polar UVI and IMAGE FUV instruments[50,51]. The onset times from different event lists were merged, and the potential existence of AKR precursors was examined prior to any event listed in the merged timeline.

Data from the Polar PWI covering the period from March 25, 1996, to September 16, 1997, with a frequency range of 13 kHz to 807 kHz, were primarily utilized for identifying AKR precursors due to their high temporal resolution (~2 seconds). Prior to this study, the existence of AKR precursors and their specific characteristics were largely unknown, with no clear reports or established criteria for identifying them. To address this gap, we conducted a manual examination of spectrograms around substorm onset, searching for consistent patterns in AKR emissions. Through this process, we discovered distinct frequency-drifting features and isolated these emissions by identifying their highest and lowest frequencies, as well as their start and end times. Given the complexity of substorm onset observations, these signals are often obscured by background emissions or appear as weak precursors. Consequently, the four key parameters—frequency range (lowest & highest frequencies), start time, and breakup time—were determined interactively by adjusting the spectrogram's color saturation for each event. This approach enabled us to enhance the visibility of weak precursor signals, providing clear evidence of their existence and characteristics despite the challenges posed by background noise. The final criteria for identifying AKR precursors are summarized as follows:

1. Timing: precursor emissions are expected to occur up to 20 min before substorm onset, in accordance with findings from superposed epoch analysis[32].

2.  Signal characteristics: The emissions should either exhibit frequency-drifting tones or show a sudden intensity enhancement in the background emissions, preceding the AKR breakup.

3.  Frequency range: The emissions are possibly more commonly observed at frequencies above 100 kHz[31,32]. However, the criteria should not be strictly constrained to this range if the first two conditions are met.

4.  Pseudo-storm and regular substorm AKR precursors: AKR precursors with frequency-drifting features often appear as multiple groups of emissions preceding substorm onset. The final group immediately preceding the onset is classified as a regular substorm precursor, while any earlier groups, if present, are categorized as pseudo-storm-related AKR precursors.

A total of 266 cases of substorm-related and 93 pseudo-storm AKR precursors were identified, as summarized in Supplementary Data 1. The rough durations (start times and end times) and frequency range (lower and upper frequency limits) of each AKR precursor were first identified from the radio spectrogram according to the empirical criterion above. Then the accurate timings and frequencies were obtained by eliminating the weak emissions within the corresponding time-frequency range with an intensity threshold of $10^{-16}$ V$^2$ m$^{-2}$ Hz$^{-1}$. The obtained time and frequencies were then used for Fig. 5a, b.

WAs near the substorm onset were identified by checking ground-based auroral images in combination with the substorm onset times. The aurora images are from the MIRACLE network cameras[68], spanning from December 1996 to December 2007. The images, stored in 8-bit grayscale JPG format, have a size of 512 × 512 pixels, corresponding to an average spatial resolution of ~1 km at an ionospheric height of 110 km. This analysis focused on auroral images captured at 557.7 nm with a cadence of ~20 seconds. Observations were collected from five stations: Abisko (ABK) at geographic latitude 68.36°N and geographic longitude 18.82°E, Muonio (MUO) at geographic latitude 68.02°N and geographic longitude 23.53°E, Kevo (KEV) at geographic latitude 69.76°N and geographic longitude 27.01°E, Kilpisjärvi (KIL) at geographic latitude 69.02°N and geographic longitude 20.87°E, and Sodankylä (SOD) at geographic latitude 67.42°N and geographic longitude 26.39°E. We define the onset time of WAs as the first appearance of periodic luminosity structures within the FOV of the all-sky cameras. The auroral breakup time is marked by the initiation of poleward expansion, consistent with established criteria for substorm onset. In events featuring multiple groups of auroral activity, earlier structures that subsequently fade or are replaced by newly formed auroral features leading directly to substorm onset are classified as pseudo-breakups.

A total of 109 substorm-related WAs events and 39 pseudo-storm-related events were identified, as summarized in Supplementary Data 2. For the duration distribution analysis, we additionally include 111 auroral bead events reported in a previous study[20].

We systematically examined all events in the MIRACLE wave-like aurora database and identified only one event that coincides both temporally and spatially with AKR precursor signatures observed by the Polar spacecraft (Supplementary Fig. 3). The case shown in Fig. 2 was independently identified by combining observations from the Arase spacecraft[56-59] and ground-based all-sky imagers from the THEMIS network[82]. However, due to limitations in Arase's orbital geometry and wave instrument configuration, such simultaneous observations are rare. The event in Fig. 3 was selected through a targeted comparison between MIRACLE auroral events and Geotail PWI data[63]. While Geotail's broader temporal coverage allowed for more overlaps with ground-based events, its limited time-frequency resolution, as shown in Fig. 3, is not optimal for resolving the fine-scale drifting features typical of AKR precursors. Consequently, all frequency-drifting analyses and statistical results presented in this study—aside from the simultaneous cases in Figs. 2 and 3—are based exclusively on Polar PWI observations.

## Tracing the frequency-drifting tones

The frequency-drifting tones in the AKR precursors were manually traced, as illustrated in Fig. 7. Only tones with sufficiently distinct features were included, resulting in a total of 686 emission tones being traced. By "sufficiently distinct," we refer to tones that exhibit clear frequency drift patterns on the spectrogram, where the intensity along the drift path is notably stronger than at the edges. A second-order polynomial fit was applied to these traced time and frequencies (i.e., $x$ and $y$ values on the spectrogram) to calculate the drift rate (df/dt) values as shown in Fig. 7b. The frequency range and duration of each drifting tone were then extracted and used in Fig. 5c.

The periods (time spacings) between drifting tones were derived by selecting adjacent emission tones. Only those tones exhibiting a distinct grouping with relatively stable periods were included in this analysis (as shown in Fig. 7c). By "distinct grouping with relatively stable periods," we refer to adjacent tones that share similar frequency drift patterns and yield consistent period calculations across multiple neighboring tones. Periods derived with sudden, significant increases compared to those of the adjacent tones were excluded. Tones lacking clear repetitive features, such as those depicted in Fig. 7e (tone-1 and tone-2), were excluded from the period analysis. These periods were averaged over all frequencies to provide one average period value for each pair of drifting tones, and these values were subsequently used in the analysis presented in Fig. 5f. The averaged df/dt values are also obtained by averaging the df/dt values in the adjacent emission tones, as shown by Fig. 7d. Both the df/dt values from each frequency-drifting tone and the averaged values are used in the analysis of Fig. 5e.

The derived frequencies and df/dt values for these frequency-drifting tones were converted to source altitudes and velocities along the magnetic field lines using the International Geomagnetic Reference Field (IGRF) model[69]. The magnetic field lines were traced using parameters from wave-like aurora forms, with the footprint connected to the average location of the MIRACLE stations at a geographic latitude of 68.51 degrees. The AKR frequencies were mapped along the magnetic field line, assuming that the emission frequencies are equal to the local electron cyclotron frequencies ($f_{ce} \approx \frac{1}{2\pi} \frac{qB}{m_e}$, where $q$ and $m_e$ are the electron charge and mass, respectively, and $B$ is the magnetic field strength). By substituting the physical constants and expressing B in units of nanotesla (nT), this relation reduces to the practical expression. The traveling distance of each frequency-drifting tone was integrated along the magnetic field line from the minimum to the maximum frequency of the tone. Note that the Tsyganenko-96 magnetic field model[83] was also tested, yielding consistent results, which is due to the fact that the AKR source regions are predominantly located close to Earth, within a radial distance range of ~1.3 to 2.4 Re.

## Calculation of the characteristic velocities in AKR source region

The Alfvén speed ($\frac{B}{\sqrt{\rho\mu_0}}$, where $B$ is the local magnetic field, $\rho$ is the mass density, and $\mu_0$ is the permeability of free space) was calculated using the IGRF magnetic field model in combination with a global electron density model[84] provided in a previous work. The ion acoustic speed ($\sqrt{\frac{T_e}{m_i}}$, where $T_e$ is the electron temperature and $m_i$ is the ion mass) was calculated independently for protons and oxygen ions, assuming average electron temperatures ranging from 0.3 to 30 electron volts (eV), given that the electron temperature remains relatively constant at AKR source altitudes, with an average value of about 3 eV[84].

## Simulation of the visible AKR sources

To theoretically determine the locations of AKR sources visible to a spacecraft at a given position, we performed simulations assuming a distribution of source regions spanning the full MLT range from 00:00 to 23:59 and L-shells from $L = 4.8-9.0$, in increments of 0.1. This range corresponds to invariant latitudes between 62° and 70°, broadly encompassing the auroral oval position. A dipole magnetic field model, with a surface field strength of −0.31 Gauss, was used to

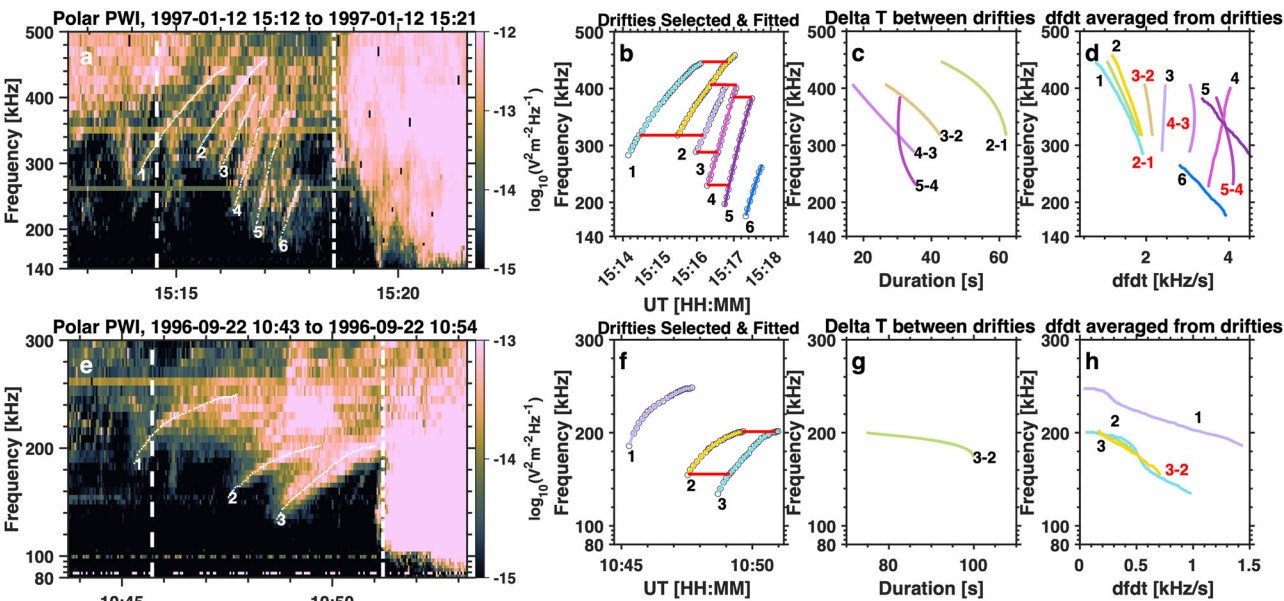

**Fig. 7 | Illustrations of traced AKR frequency-drifting tones. a** Spectrogram of the wave electric field from Polar PWI. The time range of AKR precursor emissions is indicated by the two vertical white lines. The traced frequency-drifting tones are indicated by white lines and corresponding numbers. **b** Traced frequency-drifting tones plotted as a function of time and frequency. Circles mark the selected frequency points from the original spectrogram, while colored lines represent the fitted results obtained using a second-order polynomial fit. Horizontal red lines indicate the tones used for the calculation of periods. **c** Derived periods between

frequency-drifting tones plotted as a function of duration and frequency. The label "2–1" signifies the period calculated as the time difference between the second and first tones, as indicated by the numbers in (**a**, **b**). **d** Averaged frequency drift (df/dt) values plotted as a function of df/dt and frequency (with red label "2–1"). The frequency drift rate from each single frequency-drifting tone is plotted and marked by black label text. **e**–**h** present a second example in the same format as a-d. Source data are provided as a Source Data file.

estimate the local electron cyclotron frequency. Under the assumption that AKR is generated where the emission frequency matches the local electron cyclotron frequency, the corresponding source locations along magnetic field lines were derived for each input frequency.

The simulated frequency ranges were chosen to match observed AKR precursor bands: 300–530 kHz for Fig. 2, 100–550 kHz for Fig. 3, and 100–230 kHz for Supplementary Fig. 3, with a frequency resolution of 1 kHz. For each AKR source along the magnetic field line, we determined the angle between the vector connecting the spacecraft and the source and the magnetic field direction at the source location. Spacecraft positions were set based on the actual observational intervals.

Radio visibility was defined such that the angle between the wave propagation direction (from source to spacecraft) and the magnetic field at the source matched a presumed beaming angle. While AKR is theoretically emitted perpendicular to the local magnetic field, near-source refraction typically reduces this angle to below 90°. We therefore considered beaming angles from 90° to 40°, with a hollow cone thickness of 5°, consistent with a previous study[85]. For each case, we identified all AKR sources visible to the spacecraft under these geometric constraints. The resulting distributions of visible AKR sources for the different events are shown in Figs. 2, 3, and Supplementary Fig. 3. This simulation method follows the same approach as used in previous work[86].

## Data availability

The Polar PWI data are obtained from CDAWeb at https://cdaweb.gsfc.nasa.gov/pub/data/polar/pwi/pwi_h1/. The Polar UVI auroral images from the Polar UVI instrument can be accessed at https://cdaweb.gsfc.nasa.gov/pub/data/polar/uvi/uvi_level1/. The IMAGE S12 global aurora images are obtained from the Cluster Science Archive at https://csa.esac.esa.int/csa-web/. The ground-based auroral images from Themis-ASI can be accessed at: https://themis.ssl.berkeley.edu/gmag/asi_list.php?selyear=2024&selmonth=10&selsort=ccode&smap=on&sinfo=on&seltxt=3. The geomagnetic indices are obtained at CDAWeb at https://cdaweb.gsfc.nasa.gov/pub/data/omni/omni_cdaweb/hro_1min/. Science data of the ERG (Arase) satellite were obtained from the ERG Science Center operated by ISAS/JAXA and ISEE/Nagoya University (https://ergsc.isee.nagoya-u.ac.jp/index.shtml.en, and https://ergsc.isee.nagoya-u.ac.jp/data/ergsc/satellite/erg/pwe/hfa/l2/)[58,59]. All identified AKR precursor events, along with the wave-like aurora events (including processed auroral images and corresponding movies), are available on Zenodo at: https://zenodo.org/records/16738020. The original ground-based aurora images for the identified events in this work are provided with full-resolution image data that can be accessed at (https://etsin.fairdata.fi/dataset/c2d84603-f382-4f22-ae4f-d02b7b33bea7)[87]. The Geotail PWI data illustrated in the manuscript are provided in the Source Data file accompanying this paper, while the complete dataset can be obtained upon request by contacting kojima.hirotsugu.6 m@kyoto-u.ac.jp. The event lists of AKR precursors and identified wave-like aurora are provided in the Supplementary Data file. Source data are provided with this paper.

## Code availability

The visualization and analysis of the spacecraft data and auroral images can be independently reproduced using any general-purpose programming language (e.g., Python, MATLAB, or Fortran). The simulation of radio emission visibility can be reproduced based on the descriptions provided in the Methods section.

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

## Acknowledgements

S.Y.W. and D.K.W. were supported by the Science and Technology Facilities Council (STFC) of the UK under grant ST/Y002121/1. J.E.W.'s work at LAM was supported by the Center National d'Études Spatiales (CNES). L.L. and P.Z. were supported by CNES and CNRS/INSU programs of planetology (PNP) and heliophysics (ATST, also funded by CEA). C.M.J. and A.R.F.'s work at DIAS was supported by Taighde Éireann - Research Ireland award 18/FRL/6199. The GUVI data used here are provided through support from the NASA MO&DA program. The GUVI instrument was designed and built by The Aerospace Corporation and The Johns Hopkins University. The Principal Investigator is Dr. Larry J. Paxton. S.Y.W. thanks the helpful discussion of R. Scheriber at the Space Research Center of the Polish Academy of Sciences, and Kaijun Liu at

Southern University of Science and Technology, China. S.Y.W. thanks Kirsti Kauristie at Finnish Meteorological Institute (FMI), Helsinki, Finland, and Tero Raita at Sodankylä Geophysical Observatory (SGO), University of Oulu, Finland, for providing auroral data for this study. FMI and SGO are acknowledged for operating and maintaining the ASC network, which supports long-term availability of auroral observations.

## Author contributions

Conceptualization: D.W., S.Y.W.; data calibration and technology debugging: Y. Kasaba, S.K., H.K., J.E.W., N.K., S.B.M., L.J.P., Y. Kasahara, Y.M., A.S., F.T.; methodology: D.W., S.Y.W.; Investigation: S.Y.W.; Visualization: S.Y.W.; Funding acquisition: D.W.; Project administration: D.W.; Supervision: D.W., S.Y.Y., M.M.W.; Writing—original draft: S.Y.W.; Writing—review & editing: S.Y.W., D.W., M.M.W., S.Y.Y., L.L., P.Z., C.M.J., A.R.F., J.E.W., S.B.M., Y. Kasaba, S.K., H.K., N.K., L.J.P., Y. Kasahara, Y.M., A.S., F.T.

## Competing interests

The authors declare no competing interests.

## Additional information

S. Y. Wu.

