## [Transparent Peer Review file · Nature Communications]

Radio emissions reveal Alfvénic activity and electron acceleration prior to substorm onset

Corresponding Author: Dr Siyuan Wu

Version 1:

Reviewer comments:

Reviewer #1

(Remarks to the Author)

The paper discusses the possible role of AR as precursors (or indicators of precursors) to magnetospheric substorms, by associating them with auroral beads, which are an optical phenomena associated with many substorm onsets.

This type of wave activity associated with aurora is also observed at other planets, including Jupiter and Saturn. Hence, the topic is of generally broad appeal.

There is extensive discussion on Alfvén waves and double layers, and proposes a mechanism for the frequency-drifting radio emissions that have been observed. I found this part of the manuscript speculative, but I had not major issues with it.

The entire premise of the paper hinges on the timing (and to some extent, definition) of substorm onset. In this review we do not need to get into the precise controversies surrounding this timing. However, to my eye what has been identified as a 'precursor' to substorm onset appears to be a pseudo-breakup. This is quite an important distinction, as it represents the difference between a direct and traceable 'precursor' to expansion - what the paper argues for - vs. a failed expansion (pseudo breakup). Below I lay out why this referee believes this to be the case, using some of the examples in the paper.

At this time, I cannot recommend the paper for publication. I think the paper needs to demonstrate at least the following 2 items:

- 1) The 'beads' are really beads. In at least two case (the last figure, and 3rd from last) I don't believe those would commonly be called onset beads.
- 2) That these represent pre-onset phenomena, rather than pseudo-breakup phenomena, which are often considered very different things in terms of instability growth leading to substorm expansion.

Figure 2, which is a very nice figure and very well done, highlights the issue. It appears that between 20:31 and 20:32 the beads appeared in the ASI. This activity expanded ~slightly~ then died down so that at 20:37 there was not much auroral activity. A second expansion began at around 20:38, and this expansion then led to a full breakup. Figure 2d shows the time evolution of the intensity of the AKR and the auroral brightness from the ASI. And they match almost perfectly, indicating a one-to-one association between AKR and aurora (which is already known). Note the brief local maximum at 20:34 associated with, what I argue, is the pseudo-breakup, followed by a decrease, then the full expansion a few minutes later.

Auroral beads generally last 1-2 minutes before the larger expansion. We see that here in the 20:32 and 20:33 images. The features at 20:36 and 20:37 would not generally be considered beads - those are likely auroral vortex structures that appear after initial expansion. The features identified as 'beads' at 20:38 and 20:39 would not, I believe, be called auroral beads in the standard sense.

The AKR spectrogram in Figure 2a is additionally consistent with a pseudo-breakup followed by a brief lull, then a full expansion.

There are additional examples at the end of the file. The final image is also instructive. I don't see a date on the Figure, but it's the very last one, with Geotail, Kevo and IMAGE, with an onset near 00:17. The Kevo imagery does not show auroral beads in any conventional sense. In looking at the IMAGE images it seems clear Kevo is at the Eastward edge of low level,

but sustained, auroral activity just to its West, that eventually expand Eastward to fill the FoV of Kevo. The example 2 figures earlier (no page number, but it's with the onset near 21:30) shows very similar features. This appears to be auroral activity moving from the West. These again would not be called beads, but are often observed during expansion.

The penultimate figure is quite interesting because it shows something closer to a traditional auroral bead evolution leading to breakup; although if higher temporal resolution is available I suspect it will show the auroral flowing in from the West. It does not show any AKR precursor though. This event seems to run counter to the papers conclusions.

Reviewer #2

(Remarks to the Author)

The authors suggest a connection between structured AKR emissions and auroral beads as precursors to substorm onset. They interpret the structure in the precursor AKR emissions as signatures of dispersive Alfvén waves, which have been linked to auroral beads, thereby concluding that dispersive Alfvén waves play a role in triggering substorm onset.

I would argue that the authors do not present simultaneous observations of auroral beads and structured AKR precursors. Instead, they do present AKR precursors from the low-resolution instrument on board Geotail (without discernable structure) and simultaneous auroral bead observations. Then they argue that the instrument on board Polar typically observes structured AKR precursors (because it produces high resolution spectra), and since Geotail and Polar AKR observations look similar, any precursor AKR observed by Geotail should contain similar structures (but cannot be observed directly because of the low-resolution spectra Geotail produces).

While I understand this circuitous argument (necessitated by the absence of simultaneous PWI AKR and MIRACLE auroral bead observations), it does present a certain leap of faith. In the abstract the authors write: "Simultaneous observations and statistical analysis show that both AKR precursors and auroral beads occur simultaneously [...]" I find "show" a strong word in this context, given the chain of argument outlined above. "Suggest" might be better.

Overall, I find the results compelling and worthy of publication, but I do need to see a little more analysis before I am fully convinced.

1) I am missing a discussion of visibility. The authors present (for example in Figure 2) simultaneous observations of AKR precursors and auroral beads. I note that the Geotail satellite is at about 5 MLT, -25 MLAT, and about 17 Re from Earth - where would the AKR source region be, and would it be visible from the Geotail satellite? Can you exclude other possible source locations? Since Geotail is below the magnetic equator, could the source region be from the conjugate point? Is there reason to believe that auroral beads occur in both hemispheres (along the same magnetic field line) simultaneously?

1.1) Along the same line of thought about visibility: The authors do not find any cases of simultaneous AKR precursors observed by Polar and auroral beads. Should they have found cases? Are there cases of auroral beads where Polar is in a location from which it should have seen AKR emissions? Or is Polar always in an unfavourable spot (which could be so since the authors only list about 10 cases of auroral beads during the Polar PWI data availability)?

1.2) The authors argue that Geotail precursor AKR observations are similar to Polar precursor AKR observations, and since most Polar precursor AKR show rising frequency notes, the Geotail precursor observations should probably also (although the Geotail instrument is not good enough to resolve them). Again, what is the geometry during these conjunctions (shown in Figure S2), and could the satellites observe the same source region?

2) Regarding the simultaneous Geotail/Polar observations: Some additional words about the spectra would be nice. What are these slanted lines in the Geotail spectra? Why are the PWI spectra sometimes really clear (Figure S2 panel d), and sometimes riddled by horizontal lines (Figure S2 panel h)? If the observations are from the same source, do the intensities of the observed radiation match?

Once these issues have been addressed, I support the publication of this manuscript.

Reviewer #3

(Remarks to the Author)

- What are the noteworthy results?

This is an interesting, well written and substantial manuscript highlighting the relationship of Auroral Kilometric Radiation (AKR), Alfvénic activity, and electron acceleration prior to substorm onset. These findings are of importance as they indicate that Alfvénic acceleration is potentially responsible for both optical and radio auroral features, as discussed in the paper. The authors further provide convincing evidence for the "frequency-drifting emission" features that are observed at Earth and from other magnetized planets.

Some minor comments:

The authors state in the first few sentences, "Magnetospheric substorms are among the most dynamic phenomena in Earth's magnetosphere, yet their triggering mechanisms remain unclear."

Note: There is not much discussion on how these results directly relate to triggering.

The authors comment several times that these results might have analogs on Jupiter and Saturn. "These findings suggest that the proposed mechanism may apply to radio emissions at Saturn and Jupiter, indicating a potentially universal process driving frequency-drifting radio emissions across planetary magnetospheres."

Comment: It would be interesting and helpful if the authors could expand on such general comments. Using Earth as a laboratory and analog for exploring the aurora and radio emissions at Jupiter and Saturn is an important aspect of this paper that deserves more discussion.

- Will the work be of significance to the field and related fields? How does it compare to the established literature? If the work is not original, please provide relevant references.

The results will be of interest to the atmospheric, space, solar, and planetary science communities.

The findings presented certainly highlight the importance of Alfvénic activity in substorms. The results further suggest that Alfvénic acceleration is not only responsible for optical auroral features but also for radio emissions. As stated, "These findings highlight the importance of Alfvénic activity in substorms and suggest that Alfvénic acceleration is not only responsible for optical auroral features but also for radio emissions."

Comment: I agree with the authors statement above. However, I'm finding it difficult to differentiate the current analysis from some earlier work; see, for example, selected references below. I think the manuscript could benefit from specific comments on exactly what is new and how the present analysis fits within previous AKR/substorm observations. Is the new precursor feature the significant new result that has not been observed?

REF: Su, Y.-J., R. E. Ergun, S. T. Jones, R. J. Strangeway, C. C. Chaston, S. E. Parker, and J. L. Horwitz (2007), Generation of short-burst radiation through Alfvénic acceleration of auroral electrons, *J. Geophys. Res.*, 112, A06209, doi:10.1029/2006JA012131

REF: Su, Y.-J., L. Ma, R. E. Ergun, P. L. Pritchett, and C. W. Carlson (2008), Short-burst auroral radiations in Alfvénic acceleration regions: FAST observations, *J. Geophys. Res.*, 113, A08214, doi:10.1029/2007JA012896

FYI: The authors might not be aware of this related article, which complements the space-based observations presented with related ground-based observations.

REF: LaBelle et al. (2015), Further evidence for a connection between auroral kilometric radiation and ground-level signals measured in Antarctica, *J. Geophys. Res. SpacePhysics*, 120, 2061–2075, doi:10.1002/2014JA020977

- Does the work support the conclusions and claims, or is additional evidence needed?

The conclusions and claims are supported.

- Are there any flaws in the data analysis, interpretation and conclusions? Do these prohibit publication or require revision?

The analysis presented is excellent. My only concern is how it relates and fits within the vast AKR and substorm literature.

- Is the methodology sound? Does the work meet the expected standards in your field?

The methodology is sound and clearly described in the "Methods" section.

- Is there enough detail provided in the methods for the work to be reproduced?

Yes

Version 2:

Reviewer comments:

Reviewer #1

(Remarks to the Author)

In my previous review I had recommended that the manuscript needed to demonstrate a.) that 'beads' are really beads and b.) differentiate between pseudo breakups and true onsets. The authors have meticulously responded to both requests. In particular, the separation between pseudo onsets and onsets is welcome, as is the discussion of why this distinction may be important from the standpoint of understanding the physics of onset.

I greatly appreciate the attention to detail that went into this revision, and I'm happy to recommend publication.

Reviewer #2

(Remarks to the Author)

2nd Review of "Radio emissions reveal Alfvénic activity and electron acceleration prior to substorm onset"

I would like to thank the authors for the thorough and careful revision. As far as my own comments are concerned, I feel that they have been adequately addressed. Therefore, I recommend the manuscript be published.

One final note: There is a typo in Figure 5 of the manuscript ("drfit" instead of "drift").

Reviewer #3

(Remarks to the Author)

The authors have done an excellent job addressing my concerns as well as the constructive comments from the other reviewers. The manuscript is substantially improved and acceptable for publication at this time.

Dear Editor and Reviewers,

We appreciate your kind valuable and insightful comments! We have revised our manuscript according to your comments. Thanks for your patience to wait for our revision! The point-by-point reply and revisions are listed below (answers in blue color font).

Reviewer #1 (Remarks to the Author):

The paper discusses the possible role of AKR as precursors (or indicators of precursors) to magnetospheric substorms, by associating them with auroral beads, which are an optical phenomena associated with many substorm onsets. This type of wave activity associated with aurora is also observed at other planets, including Jupiter and Saturn. Hence, the topic is of generally broad appeal. There is extensive discussion on Alfvén waves and double layers, and proposes a mechanism for the frequency-drifting radio emissions that have been observed. I found this part of the manuscript speculative, but I had not major issues with it. We thank you for this encouraging comment. The proposed mechanism is primarily based on our observational results—particularly the consistency between the frequency-drifting rates of AKR emissions (and hence spatial drift of the AKR source) and the expected propagation speed of double layers, as well as the reappearance period of these frequency-drifting tones and the period of the aurora beads. In response to this point, we have substantially revised the manuscript to clarify the supporting evidence and improve the narrative structure, as further detailed in the responses that follow.

The entire premise of the paper hinges on the timing (and to some extent, definition) of substorm onset. In this review we do not need to get into the precise controversies surrounding this timing. However, to my eye what has been identified as a 'precursor' to substorm onset appears to be a pseudo-breakup. This is quite an important distinction, as it represents the difference between a direct and traceable 'precursor' to expansion - what the paper argues for - vs. a failed expansion (pseudo breakup). Below I lay out why this referee believes this to be the case, using some of the examples in the paper.

We thank you for this very important comment. As elaborated in the responses below, we have carefully revised the manuscript to address this key issue. Specifically, we indeed observed both AKR precursor emissions and auroral bead structures during pseudo-breakup intervals and regular substorm onsets. It is well established that auroral beads can also appear during pseudo-breakups (e.g., Kalmoni et al., 2015, doi: <https://doi.org/10.1002/2015JA021470>), and previous studies have reported no significant differences in properties such as period and wavelength between beads observed during pseudo-breakups and those during regular substorms. In our initial submission, we did not systematically distinguish pseudo-breakup events from substorm-onset auroral activity, particularly in cases where both occurred in close temporal proximity or along the same pre-existing auroral arc. This also applied to the associated AKR precursors. As a result, some events were grouped under the broader category of pre-onset activity, which led to overestimated event durations. In the revised manuscript, these cases have now been reclassified accordingly.

To more clearly differentiate between the two categories, we re-examined our original event list—now expanded with additional events—using ground-based auroral imagery. When a clear dimming or disappearance separates two successive auroral activations, or when two distinct groups of

auroral structures are observed, we classify the earlier group as a pseudo-breakup (typically without or with only weak poleward expansion) and the latter group as corresponding to the regular substorm onset.

In line with this classification, the associated AKR precursor emissions frequently appear in multiple, temporally distinct groups. In such cases, we now define the group of frequency-drifting emissions closest in time to the substorm onset as the regular substorm precursor, while earlier emissions are categorized as pseudo-storm-related AKR precursors. Additional events have been included to further support this classification.

Moreover, in the statistical analysis of event durations, we have extended our dataset to include 111 auroral bead events reported by Nishimura et al. (2016, doi: <https://doi.org/10.1002/2016JA022801>), which correspond to substorm-onset auroral beads. The inclusion of these previously published events enhances the statistical robustness of our findings and confirms their consistency.

It is worth noting that here the nature of pseudo-breakups remains an open question in the community. Some studies suggest that pseudo-breakups may fundamentally differ from substorms, while others interpret them simply as weaker or short-lived substorms (e.g., Yeoman et al., 2000, <https://doi.org/10.1007/s00585-000-0897-x>, 2000.). However, since resolving this distinction lies beyond the scope of the present work, we have chosen to just reclassify the events to provide a clearer observational context.

At this time, I cannot recommend the paper for publication. I think the paper needs to demonstrate at least the following 2 items:

- 1) The 'beads' are really beads. In at least two case (the last figure, and 3rd from last) I don't believe those would commonly be called onset beads.

Thank you very much for your insightful comment regarding the classification and interpretation of auroral beads. We fully recognize the importance of carefully defining the auroral structures discussed in this work, especially when evaluating their relationship with AKR precursors. Although the simultaneous observations are not perfect due to limitations in instrumental resolution, we have now included additional events with improved data quality. Specifically, we incorporated higher-resolution measurements from the Arase spacecraft (now presented in the newly added Figure 3), as well as a newly identified event from the Polar mission (see Fig. S3).

- (1) We now show all events more clearly:

To make the classification more transparent, we have now provided all identified events in the supplementary material, including summary plots and event movies. Each event is labeled according to whether it is associated with a pseudo-breakup or a substorm onset, and we also note whether the auroral forms exhibit “beads-like” features or “wave-like” features in the early and late evolution stages.

- (2) We removed the unclear cases and added better ones:

We agree that the examples you mentioned in the original version do not clearly qualify as typical “onset beads.” These have been removed from the entire analysis (see also discussion in the

following replies) and replaced with new cases. We include a newly identified case in the supplementary material with clearer morphology and better data quality that we identified from the revised event list (see the new Fig. S3). In addition, one new high-resolution case is now given in the main text (see the new Fig. 3). The pseudo-breakup event shown in the original Figure 2 is retained (now Fig. 4), as it provides a clear example of the temporal relationship between AKR emissions and pseudo-storm aurora.

(3) It's not always easy to tell what counts as a bead:

We also want to explain that strictly classifying these auroral forms as “beads” or “not beads” is not always straightforward due to the “morphological” definition. Several factors complicate the visual identification:

- Many events start with discrete, periodic structures (beads-like) but then evolve into elongated, twisted, or vortex-like arcs—especially near substorm onset (Babu et al., 2024, doi: <https://doi.org/10.1029/2023JA031957>).

- Projection effects and viewing geometry can distort the apparent morphology. Different scientists name them differently from the morphology, e.g., the same structure might appear as a “ray” (e.g., Nishimura et al., 2016) or “spot” (Henderson, 2009, doi: <https://doi.org/10.5194/angeo-27-2129-2009>), or wave-like (Liang et al., 2008, doi: [doi:10.1029/2008GL033666](https://doi.org/10.1029/2008GL033666)).

- Resolution limits may prevent us from resolving fine-scale periodicity, even when it is physically present. For example, Kalmoni et al. (2017, doi: <https://doi.org/10.1002/2016GL071826>) used a 2D FFT to extract periodic structures from wave-like auroral arcs that looked smooth to the eye.

In short, many structures that do not look like beads may still share the same physical origin. To better reflect these complexities, we now call them “wave-like aurora” instead. We have updated the manuscript to use the term “wave-like aurora” (WAs) instead of “auroral beads.” This more inclusive wording covers both bead-like forms and other azimuthally structured features that evolve over time but still precede substorm onset (although most of our identified cases do exhibit “beads-like” features). We also clearly indicate which events start with periodic structures. We hope these clarifications and revisions address your concern. We believe the updated terminology and improved examples make the connection between AKR precursors and pre-onset auroral forms clearer.

2) That these represent pre-onset phenomena, rather than pseudo-breakup phenomena, which are often considered very different things in terms of instability growth leading to substorm expansion.

Thank you for highlighting this critical point. We agree that distinguishing between pre-onset auroral features and pseudo-breakup phenomena is essential, as they may represent fundamentally different stages of instability development leading to substorm expansion.

In our initial submission, we did not systematically separate pseudo-breakup events from substorm-onset auroral activity, especially in cases where both occurred closely in time or along the same pre-existing auroral arc. As a result, some events may have been grouped together under the broader category of pre-onset activity.

In the revised manuscript, we have carefully re-evaluated all auroral events using ground-based

imagery. We now classify auroral activity as pseudo-breakup if there is a clear dimming, disappearance, or reformation of auroral arcs prior to the main onset. This allows us to differentiate between pseudo-storm activity and the auroral structures directly leading into the substorm expansion phase. Correspondingly, we have also revised the classification of AKR precursor emissions: the final group of frequency-drifting emissions occurring closest to substorm onset is now labeled as substorm-related, while earlier groups are designated as pseudo-breakup-related.

Figure 2, which is a very nice figure and very well done, highlights the issue. It appears that between 20:31 and 20:32 the beads appeared in the ASI. This activity expanded ~slightly~ then died down so that at 20:37 there was not much auroral activity. A second expansion began at around 20:38, and this expansion then led to a full breakup. Figure 2d shows the time evolution of the intensity of the AKR and the auroral brightness from the ASI. And they match almost perfectly, indicating a one-to-one association between AKR and aurora (which is already known). Note the brief local maximum at 20:34 associated with, what I argue, is the pseudo-breakup, followed by a decrease, then the full expansion a few minutes later. Auroral beads generally last 1-2 minutes before the larger expansion. We see that here in the 20:32 and 20:33 images. The features at 20:36 and 20:37 would not generally be considered beads - those are likely auroral vortex structures that appear after initial expansion. The features identified as 'beads' at 20:38 and 20:39 would not, I believe, be called auroral beads in the standard sense. The AKR spectrogram in Figure 2a is additionally consistent with a pseudo-breakup followed by a brief lull, then a full expansion.

We agree that this event should be more appropriately categorized as a pseudo-breakup, and we have revised the manuscript accordingly.

As shown in the updated figure, the pseudo-breakup occurs very close in time and location to the subsequent substorm onset, often along the same auroral arc, with only a brief interruption in activity between the two phases. In the original submission, this entire pre-onset interval was treated as a single event, which resulted in an overestimation of the event duration. In the revised version, we have separated pseudo-breakup-associated and substorm-associated intervals for all relevant cases. This particular event is now specifically used to illustrate the correlation between AKR emissions and auroral activity during the pseudo-breakup phase, as it presents a clear observational example.

As discussed earlier, the identification of “beads” is complicated by their temporal evolution. Around 20:36–20:37 UT, the auroral forms take on a more vortex-like structure, likely representing the later stage of bead evolution. However, during 20:33–20:35 UT, periodic, bead-like structures are clearly visible and correspond well with the timing of AKR precursor emissions. A second group of auroral structures, beginning around 20:38:20 UT, also displays clear bead-like morphology at their initial appearance. In this revised version, we have revised the selection of images (shifting away from integer-minute frames) to highlight moments where the periodicity is more distinct. These “spot”-like features rapidly evolve into wave-like arcs, as seen in the zoomed-in figure above.

We note that the identification of periodic features is often more apparent in auroral movies than in static frames. To address this, we also provide movie files for all identified events in the supplementary material to enable better assessment of the temporal behavior of the auroral forms.

Importantly, to the best of our knowledge, a clear one-to-one correspondence between individual AKR structures and optical auroral beads—or any other fine-scale auroral features—has not been demonstrated in previous studies. Earlier research has primarily focused on large-scale auroral structures such as auroral bulges (e.g., Hanasz et al., 2001, doi: <https://doi.org/10.1029/2000JA900098>), largely due to the inherent difficulty of imaging AKR source regions with sufficient spatial and temporal resolution to resolve fine-scale features. In this study, such a comparison becomes feasible because the observations are made near substorm onset, a time when both AKR intensification and the development of auroral beads occur simultaneously. These coincident signatures enable a direct comparison of their temporal and spatial dynamics. Outside of substorm onset conditions, establishing such direct correspondence is generally not possible due to the unknown source region of the radio emissions. Furthermore, simultaneous observations of AKR and small-scale auroral structures remain extremely rare, given the stringent requirements on spacecraft positioning, instrumental resolution, and favorable ground-based imaging conditions.

There are additional examples at the end of the file. The final image is also instructive. I don't see a date on the Figure, but it's the very last one, with Geotail, Kevo and IMAGE, with an onset near 00:17. The Kevo imagery does not show auroral beads in any conventional sense. In looking at the IMAGE images it seems clear Kevo is at the Eastward edge of low level, but sustained, auroral activity just to its West, that eventually expand Eastward to fill the FoV of Kevo. The example 2 figures earlier (no page number, but it's with the onset near 21:30) shows very similar features. This appears to be auroral activity moving from the West. These again would not be called beads, but are often observed during expansion. The penultimate figure is quite interesting because it shows something closer to a traditional auroral bead evolution leading to breakup; although if higher temporal resolution is available I suspect it will show the auroral flowing in from the West. It does not show any AKR precursor though. This event seems to run counter to the papers conclusions.

Thank you for pointing this out. We apologize for the omission of date information in the original figure panels. Regarding the three examples previously labeled as “auroral beads,” we acknowledge that these are not ideal cases. In the revised manuscript, we have removed these examples and replaced them with newly identified events that demonstrate much clearer AKR–aurora correlations, based on an updated event list.

A brief clarification of the three original cases is as follows:

- First case: The auroral structures likely originated as beads but had already evolved into elongated, vortex-like forms by the time of observation. This kind of morphological transition is occasionally observed in other events as well, where auroral beads evolve into twisted arcs. Despite the evolution, the auroral intensification at ~00:17 UT observed in the Kevo field of view suggests a pre-onset connection with AKR activity.

- Penultimate case: This event exhibited very weak AKR precursor signatures—an observation quite common in the Polar data, as is now provided in the supplementary material summarizing all identified events. The weak appearance illustrated is likely due to the low resolution of the Geotail PWI data and hence it is difficult to distinguish.

- Last case: The auroral activity preceding the substorm onset around 21:30 UT was ambiguous, and the classification of beads in this case is uncertain. These events, including those potentially occurring during the expansion phase (as inferred from a increase of the AL index), have now been excluded from the analysis. The AKR precursor here was faint and restricted to a narrow frequency band, again likely due to instrumental limitations.

To improve the clarity and reliability of the analysis, we have excluded these ambiguous cases and introduced new events (now presented in newly added Figure 3 and supplementary Figure S3) that provide better and visually compelling evidence of the AKR–aurora connection, including clearer pre-onset bead-like features.

Furthermore, by separating the events into substorm-associated and pseudo-breakup-associated categories, a substantial portion of the manuscript has been rewritten. New figures and more detailed explanations have been added to strengthen the narrative. We believe that the revised version presents a more rigorous and general demonstration of the connection between AKR precursors and pre-onset wave-like aurora. We sincerely thank the reviewer for their insightful and constructive feedback, which helped us improve the clarity and robustness of the manuscript.

Reviewer #2 (Remarks to the Author):

The authors suggest a connection between structured AKR emissions and auroral beads as precursors to substorm onset. They interpret the structure in the precursor AKR emissions as signatures of dispersive Alfvén waves, which have been linked to auroral beads, thereby concluding that dispersive Alfvén waves play a role in triggering substorm onset. I would argue that the authors do not present simultaneous observations of auroral beads and structured AKR precursors. Instead, they do present AKR precursors from the low-resolution instrument on board Geotail (without discernable structure) and simultaneous auroral bead observations. Then they argue that the instrument on board Polar typically observes structured AKR precursors (because it produces high resolution spectra), and since Geotail and Polar AKR observations look similar, any precursor AKR observed by Geotail should contain similar structures (but cannot be observed directly because of the low-resolution spectra Geotail produces). While I understand this circuitous argument (necessitated by the absence of simultaneous PWI AKR and MIRACLE auroral bead observations), it does present a certain leap of faith. In the abstract the authors write: "Simultaneous observations and statistical analysis show that both AKR precursors and auroral beads occur simultaneously [...]" I find "show" a strong word in this context, given the chain of argument outlined above. "Suggest" might be better. Overall, I find the results compelling and worthy of publication, but I do need to see a little more analysis before I am fully convinced.

We thank the reviewer for their positive comments. While the simultaneous observations are not perfect due to limitations in instrumental resolution, we have now added new cases with improved data quality. In particular, we incorporated higher-resolution observations from Arase (now included in the newly added Figure 3), as well as a newly identified event from the Polar spacecraft (Fig. S3).

Following the suggestion of Reviewer 1, we have revised the terminology used to describe the auroral forms. Specifically, we now refer to them as "wave-like aurora" rather than "auroral beads," as distinct bead-like structures are not always clearly observed, although we believe the forms are closely related. Additionally, we have replaced the word "show" with "suggest" in several instances to avoid overstating the strength of our conclusions.

1) I am missing a discussion of visibility. The authors present (for example in Figure 2) simultaneous observations of AKR precursors and auroral beads. I note that the Geotail satellite is at about 5 MLT, -25 MLAT, and about 17 Re from Earth - where would the AKR source region be, and would it be visible from the Geotail satellite? Can you exclude other possible source locations? Since Geotail is below the magnetic equator, could the source region be from the conjugate point? Is there reason to believe that auroral beads occur in both hemispheres (along the same magnetic field line) simultaneously?

Yes, auroral beads have been observed to exhibit conjugate behavior across hemispheres, as demonstrated in previous studies. An example is shown below, with observations from Iceland (Northern Hemisphere, left) and Syowa Station in Antarctica (Southern Hemisphere, right), reproduced from Motoba et al. (2012).

(Motoba et al., 2012, doi:10.1029/2012GL051599)

We have now added a visibility test for the AKR source regions. In this calculation, AKR sources are distributed along L-shells ranging from 4.8 to 9 and over all longitudes (0° – 359°). For each source location, we test a range of beaming angles (90° , 80° , 70° , 60° , 50° , and 40°) and compute the angle between the source-to-spacecraft direction and the local magnetic field direction. This allows us to determine whether AKR emission at a given beaming angle would be visible to the spacecraft.

As shown above and in the revised manuscript, the new Panel (E) displays the visible source locations (colored markers) and visible magnetic field lines (red for the northern hemisphere and orange for the southern hemisphere). The corresponding magnetic footprints of these visible sources are projected in Panel (F), with solid contours representing the northern hemisphere and dashed contours representing the southern hemisphere.

From this analysis, we infer that if AKR is generated within the field of view (FoV) of the ABK camera—corresponding to the auroral structures observed by ABK—then for these emissions to be subsequently detected by Geotail, two conditions must be met: (1) the associated aurora would need to evolve further toward the midnight sector, and (2) the AKR beaming angle would need to remain close to 90° . Given the spatial constraints, such a scenario is rather restrictive. In contrast, Geotail is located in the southern hemisphere, making it more favorable for detecting AKR generated from

southern conjugate aurora corresponding to ABK observations. In this case, emissions beamed at 70° – 80° would be more naturally directed toward Geotail's position. Therefore, the observed AKR is more plausibly attributed to southern hemisphere sources associated with the conjugate auroral structures seen by ABK.

Furthermore, the newly added events in the revised manuscript provide improved viewing conditions, reinforcing this interpretation.

1.1) Along the same line of thought about visibility: The authors do not find any cases of simultaneous AKR precursors observed by Polar and auroral beads. Should they have found cases? Are there cases of auroral beads where Polar is in a location from which it should have seen AKR emissions? Or is Polar always in an unfavourable spot (which could be so since the authors only list about 10 cases of auroral beads during the Polar PWI data availability)?

The MIRACLE aurora cameras started operation from late 1996 and the ground-based aurora observations are generally taken during throughout the northern hemisphere winter spring season when there is enough darkness. We found one new example with the new event list and it is given now in the supplementary material.

To find the simultaneous observation, for AKR:

- (1) Spacecraft need to be positioned at latitude >30 degrees, $R_s > \sim 5 R_e$, MLT at nightside is better as suggested by the MLT-MLat- R_e distribution in Figure S1.
- (2) Spacecraft observed the substorm onset: Spacecraft observed the substorm onset (with AKR show sudden intensity enhancement and the low-frequency extension, also we need the spectrogram to be a little bit quite, so don't have strong AKR emissions before the substorm onset, which would be the background emissions that marked the precursor features).

For aurora images:

- (1) We need to use the ground-based all-sky camera as space-based images do not give good enough spatial resolution to resolve the beads.
- (2) There are aurora camera data during the desired time (specific seasons when cameras are in darkness)
- (3) Good weather with no cloud for aurora observation.
- (4) The camera should be positioned around the substorm onset position to capture the pre-onset aurora.

Therefore, such observations are extremely rare, even though substorms themselves are common and frequently seen in both AKR and auroral datasets. We examined several spacecraft datasets in an attempt to identify suitable events, but each comes with specific limitations. For instance, the IMAGE and Wind/WAVES missions lack sufficient spectral and temporal resolution; the Van Allen Probes offer limited resolution and their orbits are generally not optimal for AKR precursor observations; Akebono provides excellent resolution, but lacks continuous coverage and most of its observations are confined to polar regions. Arase data feature high resolution, but the spacecraft's position is often unfavorable, and the actual data resolution depends on the instrument's operation mode.

1.2) The authors argue that Geotail precursor AKR observations are similar to Polar precursor

AKR observations, and since most Polar precursor AKR show rising frequency notes, the Geotail precursor observations should probably also (although the Geotail instrument is not good enough to resolve them). Again, what is the geometry during these conjunctions (shown in Figure S2), and could the satellites observe the same source region?

We have now included visibility tests for the three simultaneous AKR–aurora observation events for the new Figs. 3-4 and S3. The spacecraft positions are now provided in Figure S2. AKR visibility is plausible in both cases. However, due to the lack of simultaneous auroral imaging for these events, we are unable to further constrain the precise source regions of the observed AKR emissions.

2) Regarding the simultaneous Geotail/Polar observations: Some additional words about the spectra would be nice. What are these slanted lines in the Geotail spectra?

We have now added explanations regarding the slanted patterns observed in the spectrograms. These features primarily arise from the frequency sweep logic of the receiver, and are partly influenced by the spacecraft's spin. This applies to both the Geotail and Polar datasets. In the case of Polar, when data are collected from the “Eu” antenna (the new Fig. S2 Panel B & N)—which lies within the spacecraft's spin plane—spin-induced modulations become apparent. In contrast, data from the “Ez” antenna (oriented along the spin axis, the new Panel F & J) produce spectrograms largely free from such artifacts, as seen in Panels E and J. The horizontal line around 700 kHz in the Polar spectrogram results from onboard electromagnetic interference originating from other instruments on the spacecraft. These are now explained in the Caption of Fig. S2.

Why are the PWI spectra sometimes really clear (Figure S2 panel d), and sometimes riddled by horizontal lines (Figure S2 panel h)? If the observations are from the same source, do the intensities of the observed radiation match?

The differences in noise level and spectral appearance could result from the data being returned in different operational modes—either from a short monopole antenna or from a long dipole configuration. This distinction is further clarified by the physical characteristics of the antennas: the “Eu” dipole, lying in the spacecraft spin plane, has an effective length of 230 m ($E_u = 130$ m, $E_v = 100$ m), whereas the “Ez” dipole, aligned along the spin axis, spans only 14 m (sphere-to-sphere distance). Consequently, the noise levels differ between antenna modes, even though calibration procedures account for their respective effective lengths and response parameters.

Additionally, we note that the color scales of the spectrograms are intentionally saturated differently across panels to enhance the visibility of AKR precursors. This is now explicitly mentioned in the figure caption to avoid confusion when comparing intensities between panels.

Once these issues have been addressed, I support the publication of this manuscript.

We sincerely thank you for your positive comments and for taking the time to review our manuscript.

Reviewer #3 (Remarks to the Author):

- What are the noteworthy results?

This is an interesting, well written and substantial manuscript highlighting the relationship of Auroral Kilometric Radiation (AKR), Alfvénic activity, and electron acceleration prior to substorm onset. These findings are of importance as they indicate that Alfvénic acceleration is potentially responsible for both optical and radio auroral features, as discussed in the paper. The authors further provide convincing evidence for the “frequency-drifting emission” features that are observed at Earth and from other magnetized planets.

Thank you for highlighting the significance of our findings. We are grateful for your positive and encouraging comments.

Some minor comments:

The authors state in the first few sentences, “Magnetospheric substorms are among the most dynamic phenomena in Earth’s magnetosphere, yet their triggering mechanisms remain unclear.” Note: There is not much discussion on how these results directly relate to triggering. We have now incorporated additional discussion on substorm triggering mechanisms in the Introduction, including relevant references to existing models and observational studies.

The added paragraph: The physical mechanisms underlying substorm onset are broadly categorized into two models. The near-Earth neutral line (NENL) model suggests that magnetic reconnection initiates at distances of ~10–20 Earth Radii (Re) downtail in the plasma sheet and propagates earthward^{3,5}. Conversely, the current disruption (CD) model proposes that substorm onset is driven by near-Earth instabilities—such as ballooning or cross-tail current instabilities—which disrupt the cross-tail current and trigger expansion from inside-out^{4,7,8}. Within the CD framework, structured auroral forms—particularly auroral beads that emerge during the growth-to-expansion transition—are interpreted as ionospheric signatures of equatorial plasma instabilities, potentially mediated by kinetic Alfvén wave coupling^{9,10,11}. These auroral structures thus serve as a vital diagnostic of magnetospheric instability and energy transfer.

The authors comment several times that these results might have analogs on Jupiter and Saturn. “These findings suggest that the proposed mechanism may apply to radio emissions at Saturn and Jupiter, indicating a potentially universal process driving frequency-drifting radio emissions across planetary magnetospheres.” Comment: It would be interesting and helpful if the authors could expand on such general comments. Using Earth as a laboratory and analog for exploring the aurora and radio emissions at Jupiter and Saturn is an important aspect of this paper that deserves more discussion.

We have now expanded the discussion on this aspect to emphasize the potential applicability of the proposed mechanism in explaining frequency-drifting emissions observed at Saturn and Jupiter.

The added sentences: Similar frequency-drifting structures have also been identified in Saturn’s kilometric radiation, with drifting rates ranging from a few kHz/s to as high as ~240 kHz/s⁷². At Jupiter, periodic radio sources moving at local ion acoustic speeds have been observed in radio source regions^{73,74}. These observations infer small-scale electric potential structures that strongly suggest the presence of double layers. Moreover, the role of Alfvén waves in generating radio

emissions at Jupiter is even more prominent and well established—particularly in satellite-induced emissions^{54, 68, 73, 74}. These findings suggest the idea that the mechanism proposed in this study—where frequency-drifting AKR emissions are driven by double layers formed through dispersive Alfvén wave interaction—may also apply to Saturn and Jupiter. This points to a potentially universal process responsible for frequency-drifting radio emissions across magnetized planetary environments.

- Will the work be of significance to the field and related fields? How does it compare to the established literature? If the work is not original, please provide relevant references.

The results will be of interest to the atmospheric, space, solar, and planetary science communities.

We sincerely thank you for your positive evaluation.

The findings presented certainly highlight the importance of Alfvénic activity in substorms. The results further suggest that Alfvénic acceleration is not only responsible for optical auroral features but also for radio emissions. As stated, “These findings highlight the importance of Alfvénic activity in substorms and suggest that Alfvénic acceleration is not only responsible for optical auroral features but also for radio emissions.”

Comment: I agree with the authors statement above. However, I’m finding it difficult to differentiate the current analysis from some earlier work; see, for example, selected references below. I think the manuscript could benefit from specific comments on exactly what is new and how the present analysis fits within previous AKR/substorm observations. Is the new precursor feature the significant new result that has not been observed?

Yes, these precursor features are indeed first reported in this study. Previous works have mainly relied on superposed epoch analyses, suggesting the existence of such signals but without detailed case studies or clear observational evidence. The references below, which are now cited in the manuscript, focus primarily on finer AKR structures—such as S-burst AKR—with durations on the order of seconds. These short-timescale emissions resemble the so-called Jupiter S-bursts and are believed to be generated directly by Alfvén waves through electron acceleration and motion. These phenomena differ from our identified AKR precursors both in timescale and physical mechanism, representing different processes in AKR generation (double-layer formation by Alfvén waves as proposed in this work).

The added sentences:

It is important to note that the AKR precursor frequency-drifting tones discussed here should not be confused with previously reported fine structures of AKR, such as striated AKR69—characterized by rapid negative frequency drifts (~5 kHz/s), typically observed at frequencies below ~90 kHz—and S-burst-like AKR emissions^{44, 45}, which exhibit even faster drift rates (>10 kHz/s), shorter durations, and recurrence intervals of ~7–18 kHz. Additionally, “dot” AKR—appearing as spot-like features in AKR spectrograms and predominantly observed below 100 kHz—has also been reported⁴³. Although both S-burst-like and dot AKR emissions have been proposed to be associated with Alfvén wave activity, they are generally attributed to direct Alfvénic acceleration without invoking the formation of double layers. A further key distinction lies in their contextual association: the fine structures reported in earlier studies have not been systematically linked to specific magnetospheric processes and are more likely to represent ubiquitous features of AKR. In contrast,

the frequency-drifting AKR precursors identified in this study are consistently observed immediately prior to substorm onset or during pseudo-substorm auroral activity, indicating a different origin and potentially distinct generation mechanism.

REF: Su, Y.-J., R. E. Ergun, S. T. Jones, R. J. Strangeway, C. C. Chaston, S. E. Parker, and J. L. Horwitz (2007), Generation of short-burst radiation through Alfvénic acceleration of auroral electrons, *J. Geophys. Res.*, 112, A06209, doi:10.1029/2006JA012131

REF: Su, Y.-J., L. Ma, R. E. Ergun, P. L. Pritchett, and C. W. Carlson (2008), Short-burst auroral radiations in Alfvénic acceleration regions: FAST observations, *J. Geophys. Res.*, 113, A08214, doi:10.1029/2007JA012896

FYI: The authors might not be aware of this related article, which complements the space-based observations presented with related ground-based observations.

REF: LaBelle et al. (2015), Further evidence for a connection between auroral kilometric radiation and ground-level signals measured in Antarctica, *J. Geophys. Res. Space Physics*, 120, 2061–2075, doi:10.1002/2014JA020977

We have now incorporated these references and included relevant discussion in the revised manuscript.

- Does the work support the conclusions and claims, or is additional evidence needed?

The conclusions and claims are supported.

- Is the methodology sound? Does the work meet the expected standards in your field?

The methodology is sound and clearly described in the “Methods” section.

- Is there enough detail provided in the methods for the work to be reproduced?

Yes

We sincerely thank you for your positive evaluation.

- Are there any flaws in the data analysis, interpretation and conclusions? Do these prohibit publication or require revision? The analysis presented is excellent. My only concern is how it relates and fits within the vast AKR and substorm literature.

In response to the comments from Reviewer 1, we have reclassified the AKR emissions and auroral observations into pseudo-breakup-associated and regular substorm-associated events. Accordingly, we have expanded the introduction on these auroral and AKR phenomena and added a new Figure 1 as well as revised Figure 2 to improve the logical flow of the manuscript. More detailed explanations are now provided.

We sincerely thank you for your positive comments and for taking the time to review our manuscript.

Dear Editor and Reviewers,

We appreciate your kind valuable and insightful comments all through the peer review process!

Reviewer #1 (Remarks to the Author):

In my previous review I had recommended that the manuscript needed to demonstrate a:) that 'beads' are really beads and b) differentiate between pseudo breakups and true onsets. The authors have meticulously responded to both requests. In particular, the separation between pseudo onsets and onsets is welcome, as is the discussion of why this distinction may be important from the standpoint of understanding the physics of onset.

I greatly appreciate the attention to detail that went into this revision, and I'm happy to recommend publication.

We sincerely thank you for your positive comments and for taking the time to review our manuscript.

Reviewer #2 (Remarks to the Author):

2nd Review of "Radio emissions reveal Alfvénic activity and electron acceleration prior to substorm onset"

I would like to thank the authors for the thorough and careful revision. As far as my own comments are concerned, I feel that they have been adequately addressed. Therefore, I recommend the manuscript be published.

One final note: There is a typo in Figure 5 of the manuscript ("drfit" instead of "drift").

Done, the typo has been corrected. We sincerely thank you for your positive comments and for taking the time to review our manuscript.

Reviewer #3 (Remarks to the Author):

The authors have done an excellent job addressing my concerns as well as the constructive comments from the other reviewers. The manuscript is substantially improved and acceptable for publication at this time.

We sincerely thank you for your positive comments and for taking the time to review our manuscript.